# Barriers and Facilitators for Implementing Music Interventions in Care Homes for People with Dementia and Depression: Process Evaluation Results of the Multinational Cluster-Randomized MIDDEL Trial

**DOI:** 10.3390/bs15081004

**Published:** 2025-07-23

**Authors:** Naomi Rasing, Annemieke Vink, Mirjam Schmitz, Jo Dugstad Wake, Monika Geretsegger, Vigdis Sveinsdottir, Christian Gold, Yesim Saltik, Hazal Nevruz, Burcin Ucaner, Ulrike Frischen, Johanna Neuser, Gunter Kreutz, Joanne Ablewhite, Justine Schneider, Sytse Zuidema, Sarah Janus

**Affiliations:** 1Department of Primary and Long-Term Care, University of Groningen, University Medical Center Groningen, 9713 GZ Groningen, The Netherlands; mirjam_schmitz@hotmail.com (M.S.); s.i.m.janus@umcg.nl (S.J.); 2Alzheimer Center Groningen, University Medical Center Groningen, 9713 GZ Groningen, The Netherlands; 3ArtEZ Professorship Music-Based Therapies and Interventions, Music Therapy Department, Academy of Music, ArtEZ University of the Arts, 7511 PN Enschede, The Netherlands; a.vink@artez.nl; 4The Grieg Academy Music Therapy Research Centre, NORCE Norwegian Research Centre, 5008 Bergen, Norway; jowa@norceresearch.no (J.D.W.); moge@norceresearch.no (M.G.); chgo@norceresearch.no (C.G.); 5Health & Social Sciences, NORCE Norwegian Research Centre, 5008 Bergen, Norway; visv@norceresearch.no; 6Türk Müziği Devlet Conservatory, Hacı Bayram Veli University, 06560 Ankara, Türkiye; multicultimuziektherapeut@outlook.com (Y.S.); nevruz.hazal@hotmail.com (H.N.); burcinucaner@yahoo.com (B.U.); 7Department of Music, Carl Von Ossietzky Universität, 26129 Oldenburg, Germany; ulrike.frischen@gmail.com (U.F.); gunter.kreutz@uni-oldenburg.de (G.K.); 8Epidemiology and Biometry, Carl Von Ossietzky Universität, 26129 Oldenburg, Germany; johannaneuser@aol.com; 9Faculty of Social Sciences and Institute of Mental Health, University of Nottingham, Nottingham NG5 1PB, UK; joanne.ablewhite1@nottingham.ac.uk (J.A.); justineschneider2013@gmail.com (J.S.)

**Keywords:** long-term care, dementia, music-based intervention, music therapy, singing, program evaluation, process evaluation, implementation

## Abstract

A process evaluation was embedded in the multinational Music Interventions for Dementia and Depression in ELderly care (MIDDEL) trial to better understand barriers and facilitators for implementing music-based interventions (MBIs). Stakeholders from 66 care home units across 5 countries completed a survey at baseline (*n* = 229) and after a six-month intervention period (*n* = 101), comparing expectations and experiences between countries, intervention groups, and stakeholders. MBIs were evaluated and found to be relevant and feasible. Barriers include a lack of support, turnover among employees, and a lack of motivation. Facilitators include individual stakeholders who proactively facilitate and stimulate implementation, as well as the presence of stable, well-functioning teams, clear communication, and adhering to project plans. Fewer barriers than expected related to care staff workload and the time needed for implementing new MBIs in care homes. MBIs can be beneficial for people with dementia, yet implementation in care homes can be challenging due to contextual factors. Involving stakeholders in key positions is essential: care home managers are pivotal for policy-making and the sustainable adoption of MBIs, whereas the commitment and the involvement of care staff are needed for day-to-day implementation. Insight into these barriers to and facilitators of implementation can contribute to the interpretation of trial results.

## 1. Introduction

Over 55 million people worldwide live with some type of dementia, a degenerative syndrome that affects cognitive and behavioral functioning. This number is expected to increase in the coming decades ([6]). Depression is a common manifestation of neuropsychiatric symptoms in dementia, although the nature of the relationship between dementia and depression remains unclear ([17]; [18]; [32]). Music-based interventions (MBIs) have been found to be effective in improving symptoms of depression and quality of life in people with dementia ([3]; [13]; [30]), but the quality of evidence of these non-pharmacological interventions varies ([19]; [31]). Most studies to date that assess the effectiveness of MBIs have a small sample size, weak research design, or insufficiently reliable results ([30]). Often, a description of basic elements and the rationale for the intervention is lacking ([26]). Detailed intervention description and research reporting are essential components to be able to replicate, interpret, and translate research findings to practice ([26]). MBIs are complex and are delivered and evaluated at multiple levels ([25]). There can be interacting components, various stakeholders with different characteristics and needs, and different contexts in which the intervention is delivered. The implementation and uptake of an MBI can vary between care home organizations due to a range of organizational, staff-related, and environmental factors.

Common MBI implementation strategies for people with dementia in care homes include education, quality management, planning, restructuring, and cost. Outcomes relating to implementation include appropriateness, fidelity, adoption, sustainability, and acceptability. It is crucial to differentiate between studies focusing on music therapy provided by a trained music therapist and individualized music provided by care staff or researchers ([1]). Implementing a third type of MBI—individualized music listening—points to the relevance of contextual factors, such as the presence and influence of care staff during sessions, location of the MBI, and interaction with the environment ([28]). Barriers to implementation most frequently cited include limited staff time, low staff engagement, lack of training, high costs, and technical issues. Facilitators of implementation include stakeholder support, observed positive effects, training, family involvement, accessible equipment, and financial resources ([15]; [27]). However, evidence concerning barriers to and facilitators for MBIs remains limited.

To address this gap, a process evaluation of the multinational trial Music Interventions for Dementia and Depression in ELderly care (MIDDEL) was conducted to gain insight into implementation in various country contexts, to assess whether MBIs were implemented as intended, and to identify factors which appear to determine successful implementation. In other words, the overall aim of this study was to assess barriers and facilitators for MBI implementation across countries, care homes, and residents. Evaluation of these contextual aspects of implementation can inform the overall assessment of the acceptability and feasibility of MBIs. In accordance with Leontjevas’ framework, first- and second-order process data were examined ([16]). The first-order process evaluation includes assessment of intervention quality, internal validity, and external validity, following which intervention effects and cost effectiveness can be examined. A second-order process evaluation combines implementation knowledge, such as insights into barriers and facilitating factors, which may inform future interventions and studies.

The primary objective of the process evaluation was to gain insight into first-order process data and second-order process data. First-order process data include: intervention quality, measured by satisfaction, relevance, and feasibility. Second-order process data include: presence and influence of anticipated and experienced potential barriers and facilitators for implementation of MBIs for people with dementia and depression in care homes of different stakeholders. To this end, differences between expected and perceived barriers and facilitators between allocated intervention groups (GMT, RCS, GMT + RCS combined, control) at baseline (T0) and post-intervention (T6) were described in the current study. Differences between stakeholders and countries were described first to get a thorough picture of potential barriers and facilitators that may occur in day-to-day practice.

## 2. Materials and Methods

### 2.1. About the MIDDEL Trial

Music Interventions for Dementia and Depression in ELderly care (MIDDEL) was a multinational cluster-randomized controlled trial, conducted in Australia, Norway, Germany, the UK, Türkiye, and the Netherlands. A total of 86 care home units (CHUs, an organizational cluster of residents that live together and are cared for by care staff) with 1021 residents were included in the main study. Each CHU with its consenting residents with dementia and (at least mild) depressive symptoms was randomly allocated to one of four intervention groups: (1) group music therapy (GMT), (2) recreational choir singing (RCS), (3) the combination of both (combi), (4) care as usual (control group receiving no MBIs) (Figure 1). GMT and RCS are two widely used group MBIs ([9]). GMT was offered by a trained music therapist to groups of approximately five to seven care home residents. GMT sessions consisted of an introduction with a welcome song, singing familiar songs and reminiscence, optional improvisation on instruments, optional movement to music, singing familiar songs again, and ending with a closing song. The core principles of GMT were affected by regulation through reciprocal musicking, attunement, meeting the psychosocial needs of individual residents, working in the here and now, and relationship building. RCS was offered by a skilled musician in groups of ten or more residents. RCS sessions consisted of an introduction with a welcome song, singing familiar songs, learning new songs, and ending with a closing song. The core principles of RCS were to sing familiar songs, provide a familiar musical environment, foster connectedness, emotional well-being, and enjoyment. After completing baseline assessment (T0), MBIs were offered twice a week for 45 min per session in months 1–3, and once a week for 45 min in months 4–6. Follow-up assessments in the main study were carried out by blinded assessors at 3 (T3), 6 (T6), and 12 (T12) months after randomization. The aim of MIDDEL was to assess and compare the effectiveness of GMT and RCS, primarily on depressive symptoms in care home residents with dementia (clinical trial registration nr NCT03496675, [9]). The primary outcome was the severity of depressive symptoms at six months (T6) measured with the Montgomery-Åsberg Depression Rating Scale (MADRS) ([20]). Secondary outcomes included cognitive functioning, neuropsychological symptoms of dementia, quality of life, well-being, stress levels, care staff sick leave days, and caregiver burden.

### 2.2. Study Design

A survey was developed to assess first- and second-order process data related to the MIDDEL trial, which is described in detail elsewhere ([9]). The Leontjevas framework ([16]) has been used similarly by previous studies ([2]; [4]; [14]; [34]). This framework argues that once the validity of the study or intervention has been confirmed (step 1), effects and cost-effectiveness can be examined (step 2), which together with knowledge about implementation (step 3) can be decisive for further implementation or improvement of the intervention (Figure 2) ([16]). The focus of the current study was on intervention quality (relevance, feasibility, satisfaction) and on barriers to and facilitators for implementation. The process evaluation was embedded in the main MIDDEL trial (Figure 1) and included the five European countries involved in the project (excluding Australia due to differences in time of recruitment). The optional survey at timepoint T12 was excluded from the current study, due to the primary outcome being at T6, and there was a risk of recall bias among participants six months post-intervention period.

### 2.3. Participants

Care home managers, care staff, and interventionists associated with CHUs and residents involved in the MIDDEL trial were eligible for participation. They were asked for their views about the implementation of the MIDDEL trial. We defined care home managers as persons in a managerial role, including policy and development functions, staff officers, and supervisors. Care staff were defined as registered nurses, enrolled nurses, personal care attendants, allied health professionals, leisure staff, or others. Interventionists were music therapists and choir leaders who provided MBIs (GMT and/or RCS) at one or more participating care homes. This categorization of roles was standardized across countries, and overlap between stakeholder roles was not possible.

### 2.4. Survey Development

The survey was initially developed in English by NR, MS, SJ, AV, and SZ. To enable comparisons between countries and reduce language barriers, only closed binary questions (such as yes-no questions) and five-point Likert scale questions were used. The first version of the survey was shared with all researchers involved in the MIDDEL trial for feedback. In the adapted second version of the survey, each country marked questions that did not apply to their national situation and sent the remarks to the survey to the Dutch researchers, who adapted the survey accordingly. Next, researchers from each country translated the survey into their national language (Dutch, German, Norwegian, and Turkish). The Dutch MIDDEL Team developed a REDCap database for the process evaluation, including the Multi-Language Management module that enabled each respondent to fill out the survey in their national language ([11]). The survey was available in paper format and online via the REDCap electronic data capture web application ([11]), hosted at the independent research institute NORCE ([22]). The survey was distributed (1) at baseline (T0, prior to allocation to one of the four intervention groups), (2) six months after randomization, at the end of the intervention period (T6, post-intervention), and (3) an optional long-term follow-up twelve months after randomization (T12, follow-up). The T0 survey focused on expectations of participants about the trial, whereas the T6 and T12 surveys focused on experiences. The survey was adapted to be appropriate for the targeted stakeholders, meaning the survey had minor differences for each stakeholder: management, care staff, and interventionists (Appendix A).

Common barriers and facilitators for the implementation of complex psychosocial interventions in care homes were identified in previous RCTs and formed the basis of the survey. The survey consisted of multiple sections that were similar across stakeholders and timepoints: (1) informed consent; (2) demographics; (3a) relevance and feasibility; (3b) effect; (3c) satisfaction; (3d) implementation and sustainability; (4) barriers and facilitators related to care home organization and staff; (5) barriers and facilitators related to MBIs. The survey questionnaire for “Care staff” can be found in Appendix A (English version), as most surveys were completed by care staff.

### 2.5. Procedure

The research team in each country distributed study information and questionnaires in paper format and/or electronically via email. The aim and procedures of the process evaluation were explained in the national language, and all participants provided informed consent before answering the survey. Initially, a response rate of 30% was targeted to be able to make reliable statements about barriers and facilitators to implementation of the MBIs, and to ensure the collection of a full range of perspectives of different stakeholders from different countries. The research team in each country decided for themselves how many questionnaires to send out, depending on the burden and time available. Each country distributed the survey to different stakeholders (management, interventionists, and care staff) at 2 or 3 timepoints (T0, T6, optional T12). In practice, this meant –assuming 4 CHUs participated per care home organization—that in each care home organization in each country at timepoints T0, T6 (and optional T12), questionnaires were completed by at least: 1 care home manager; 2 interventionists (1 music therapist, 1 choir leader); a total of 4 care staff. Surveys were distributed from the start of the European MIDDEL trial in 2020 until the last assessments in summer 2023.

### 2.6. Data Analysis

Data were analyzed using SPSS version 28. Demographics, views related to relevance and feasibility, effect and satisfaction, implementation and sustainability, and barriers to and facilitators for implementation from care home managers, care staff, and interventionists of the participating countries were analyzed using descriptive statistics. The authors decided not to include a statistical test to assess whether differences between allocated groups were significant, as this would go beyond the descriptive purpose of the study. First, data were stratified by stakeholder to describe differences between stakeholders at baseline (T0) and post-intervention (T6). Second, data were stratified by country to describe baseline (T0) differences. Third, data were stratified by allocated intervention group. Sample size varied across countries, stakeholders, and timepoints. Median and interquartile ranges were provided for all Likert-scale questions. For yes/no questions, percentages were provided. Differences between stakeholders, countries, and intervention groups were visualized using heatmaps. For pragmatic reasons, due to the amount of data, differences in the percentage of respondents that answered ‘Yes’ to whether a statement would be/was applicable are described in text only if the difference within/between groups is larger than 20%. Similarly, differences in the median of the extent to which a factor influenced implementation are described only for those factors illustrating a difference of 2 points or more within/between groups.

## 3. Results

The survey was completed by 229 respondents at baseline and by 101 respondents at T6. A total of 453 surveys were started, of which 11 were excluded (6 respondents only filled out the informed consent form, 5 respondents from the UK filled out a survey not corresponding to their stakeholder role). In Section 3 of this paper, the largest observed differences between stakeholders, countries, and intervention groups are highlighted and discussed. Figure 3 provides an overview of the total number of completed surveys (*n* = 442 surveys) during the MIDDEL trial, of which 330 (T0 plus T6) were included in the current study.

### 3.1. Comparison at Stakeholder-Level

Baseline and post-intervention surveys were filled out by care home managers (*n* = 36 at baseline, *n* = 19 post-intervention), interventionists (*n* = 32 at baseline, *n* = 19 post-intervention), and care staff (*n* = 161 at baseline, *n* = 65 post-intervention). Descriptive characteristics of the sample at baseline (T0) and post-intervention (T6) are presented per stakeholder group (Table 1). A majority of respondents were care staff, and in each stakeholder group, most were female (72.7–88.9%). A majority of interventionists were aged 18–29 years (38.5–50.0%). Over 50% of care staff had more than a decade of work experience, while around one-third of interventionists had less than two years, and over half of them had less than five years of work experience. Less than one third (*n* = 7, 28.0%) of interventionists worked at the care home organization before the start of the MIDDEL trial. Twenty percent of care home management and care staff indicated that no MBI, such as GMT or RCS, was being provided in their care home. In each stakeholder group, the majority of respondents were involved throughout the duration of the MIDDEL trial. Stakeholders were not blinded to intervention group allocation. Nevertheless, at post-intervention (T6), half of the stakeholders did not know the exact intervention group (GMT, RCS, COMBI, or CONTROL) their CHU(s) to which their residents were allocated.

Expectations (T0) and experiences (T6) regarding relevance and feasibility were comparable across stakeholder groups, except for the statement ‘I know which tasks I have to fulfill for the MIDDEL-project’ at baseline, interventionists strongly agreed (median 5.0), most care staff neither agreed nor disagreed (median 3.0) (Table 2). Across stakeholders, the effect of MBIs was noticed most in care home residents. Interventionists reported the highest percentage of observed effects, followed by managers.

Heatmaps are used to show differences within and between stakeholders before (T0) and after (T6) the intervention period of six months from the stakeholder (Figure 4), country (Figure 5), and intervention group (Figure 6) perspectives. Grey columns represent the applicability of barriers and facilitators, with darker grey indicating more respondents expected/experienced that factor to be present. Blue columns represent the influence of those barriers and facilitators, with darker blue indicating higher expected/experienced influence of a factor.

Amongst interventionists, the barrier of ‘lack of support’ was cited more than expected. In contrast, the barriers ‘high workload of care staff’ and ‘keeping a music intervention running’ were less present than expected. The facilitator’s ‘people involved facilitating and stimulating implementation’ was cited less than expected by interventionists. The barrier ‘lack of care staff’ was mentioned less than expected, both by care staff and by interventionists. For care staff, the barriers ‘reorganizations’, ‘participating in other projects’, and ‘general resistance to change’ were less frequent than expected. The barrier, ‘in general, implementing an intervention on the CHU takes a lot of time’ was mentioned less than expected by all stakeholder groups.

Differences between stakeholders’ expectations at baseline show that most care home managers expected the facilitator’s ‘open communication and flat hierarchy’ to be high, and the barriers ‘lack of care staff’ and ‘reorganizations’ to be low, in contrast to the expectations of care staff and interventionists. Managers expected the facilitator’s ‘integration of music interventions in the care program’ to be present more than interventionists did. Compared to care staff, more managers anticipated ‘difficulties due to COVID’. Interventionists expected the barriers ‘participating in other projects’ and ‘changes in employees’, more than care staff and management did. Compared to care staff, interventionists had lower expectations of ‘lack of support’ and ‘general resistance to change’. Compared to managers, more interventionists expected there to be a ‘high workload for care staff’.

Differences between stakeholder experiences of the MBIs indicate that the presence of the facilitator’s ‘stable, well-functioning teams’ was lower, while the barriers ‘changes in employees’ and ‘lack of motivation among those involved’ were more common for interventionists. Compared to care home managers, interventionists experienced more ‘lack of support’, but less presence of ‘participating in other projects.’ However, the facilitator’s ‘people involved facilitating and stimulating implementation’ was more frequent for interventionists than managers.

The influence of barriers and facilitators differed within stakeholder groups (2.0 points difference in median) for three factors: influence of the facilitator ‘stable and well-functioning teams’ was larger than expected according to interventionists (median 2.0 to 4.0), whereas influence of the barrier ‘lack of motivation amongst those involved’ was lower than expected (median 4.0 to 2.0). For care home managers, the influence of the barrier, ‘in general, implementing an intervention on the CHU takes a lot of time’, was lower than expected (median 4.0 to 2.0).

Differences between stakeholders in expected influence of factors indicated that interventionists expected less influence of the facilitator’s ‘stable, well-functioning teams’ (median 2.0 vs. 4.0) and the barrier ‘participating in other projects’ (median 1.0 vs. 3.0) than care home managers.

Differences between stakeholders in experienced influence of factors indicate that care home managers experienced higher influence of the facilitator ‘open communication and flat hierarchy’ compared to interventionists (median 5.0 vs. 3.0), and less influence of the barrier ‘reorganizations’ compared to care staff (median 1.0 vs. 3.0).

### 3.2. Comparison at Country-Level

For the comparison between countries, baseline (T0) surveys from the Netherlands (*n* = 93), Norway (*n* = 49), Türkiye (*n* = 61), and Germany (*n* = 24) were included. Surveys from the United Kingdom (*n* = 2) were excluded. Countries’ expectations (T0) about relevance and feasibility were comparable, except for the statement ‘I know which tasks I have to fulfill for the MIDDEL-project’, where German respondents strongly agreed (median 5.0) and most Dutch respondents neither agreed nor disagreed (median 3.0) (Table 3).

Expectations about the presence of barriers and facilitators varied greatly between countries, with percentages differing up to 100% (Figure 5). Across countries, the expected presence of factors was highest for organizational barriers ‘high workload’ and ‘lack of care staff’, whereas the expected presence of the barrier ‘lack of support’ was low. Expected presence of factors related to MBI implementation was highest for facilitators ‘clear communication’, ‘people involved facilitate implementation’, and ‘project planning is maintained’ (Figure 5). In 8 out of 9 organizational factors, the percentage in Germany was highest for facilitators and lowest for the expected presence of hindering factors. For Türkiye expected presence was highest in 3 organizational barriers and lowest for the barrier ‘implementing an intervention on the CHU takes a lot of time’, compared to the other countries. Of 8 factors related to MBI implementation, the expected presence of the barrier ‘keeping a music intervention running is a lot of work’ was much lower for Türkiye, whereas the barriers ‘lack of motivation’ and ‘scarce resources’ were higher than in other countries. Expected presence of the barrier ‘changes in employees’ was lower in Norway compared to the other countries.

Across all countries, the expected influence of factors affecting implementation of the MIDDEL trial and its MBIs was largest for the organizational barriers ‘high workload’, ‘lack of care staff’, and the facilitators ‘open communication and flat hierarchy’ and ‘well-functioning teams’. In each country, the same three facilitators related to the implementation of MBI had the largest expected influence: ‘clear communication’, ‘people involved facilitate implementation’, and ‘planning is maintained’ (Figure 5).

Between countries, differences in median expected influence were larger for organizational factors than for factors related to MBI implementation. For 5 out of 9 organizational factors, Türkiye had a higher median expected influence compared to some of the other countries (a difference of ≥2.0 points on the median). This applied to the facilitator’s ‘open communication and flat hierarchy’, and the barriers ‘lack of support’, ‘reorganizations’, ‘participating in other projects’, and ‘general resistance to change’.

### 3.3. Comparison at Intervention Group-Level

For analysis of differences between intervention groups, twenty-five surveys were excluded: 22 surveys were from respondents involved in multiple intervention groups, and for 3 surveys, the allocated intervention group was unknown. The number of respondents varies per survey item, as no items were compulsory. Most respondents were in the COMBI group (*n* = 39 at baseline, *n* = 32 post-intervention), followed by CONTROL (*n* = 30 at baseline, *n* = 25 post-intervention), GMT (*n* = 26 at baseline, *n* = 23 post-intervention), and RCS (*n* = 24 at baseline, *n* = 17 post-intervention).

Between all four intervention groups (GMT, RCS, COMBI, and CONTROL), expectations (T0) regarding relevance and feasibility were comparable, except for a median difference of 1.5 points for the statement ‘participating in the project will be relevant for this care home organization’: care home managers in the RCS arm did not agree nor disagree (median 3.0), in the COMBI arm they agreed (median 4.0), and in the CONTROL and GMT arm they agreed—strongly agreed (median 4.5) (Table 4). At post-intervention, care home managers from the GMT group strongly agreed that ‘participating in the project was relevant’, whereas managers from the CONTROL group neither agreed nor disagreed.

Respondents were asked whether they noticed any effect of the MBIs (in residents, care staff, or on the CHU). In the RCS arm, almost all respondents noticed an effect in residents (93.3%). In the GMT and in the COMBI arm, more than half of the respondents noticed an effect in residents (60.0% and 57.1%, respectively). Notably, almost half of the respondents in the CONTROL arm also reported that they noticed an effect of the MBI on care home residents (28.0%), care staff (8.0%), or on the CHU (12.0%), even though they received no MBI as part of the MIDDEL trial.

For all intervention groups, perceived presence of the barriers ‘lack of care staff’ and ‘in general, implementing an intervention on the CHU takes a lot of time’ was less frequent (difference ≥20%) than expected (Figure 6). Within GMT, organizational barriers, ‘lack of support’, ‘reorganizations’, ‘participating in other projects’, and ‘resistance to change’ were cited less than expected. Related to MBI implementation, the perceived presence of the barriers ‘keeping a music intervention running is a lot of work’ and ‘difficulties due to COVID’ was lower than expected. Within RCS, the organizational barriers ‘resistance to change’, ‘high workload for care staff’, and barriers related to MBI implementation, ‘keeping a music intervention running is a lot of work’, and ‘difficulties due to COVID’ were noted less than expected. Within COMBI, the organizational facilitator ‘stable and well-functioning teams’ was cited more than expected, whereas the facilitator ‘clear communication’ was cited less than expected. Within the CONTROL group, the organizational facilitator ‘stable and well-functioning teams’ was mentioned by stakeholders more than expected, while the organizational barriers ‘reorganizations’ and ‘high workload of care staff’ were less frequent than expected. Related to MBI implementation, the barrier ‘lack of motivation’ was noted less than expected.

In RCS at baseline, between groups, in 4 out of 9 organizational factors, expectations were highest for the factors ‘open communication and flat hierarchy’ and ‘high workload for care staff’, and lowest for ‘participating in other projects’ and ‘lack of care staff’. In GMT, for 7 out of 10 factors related to MBI implementation, expectations were highest for the barriers ‘keeping an intervention running is a lot of work’, ‘changes in employees’, ‘difficulties due to COVID’, and ‘in general, implementing an intervention on the CHU takes a lot of time’, but lower compared to other groups for the facilitators ‘project planning is maintained’, ‘clear communication’, and ‘there are musical activities in the care home’. In 6 out of 10 factors, COMBI had the lowest score for expected barrier ‘keeping a music intervention running is a lot of work’, but the highest score for the barrier ‘in general, implementing an intervention on the CHU takes a lot of time’ as well as for the facilitators ‘project planning maintained’, clear communication’, ‘there are musical activities in the care home’, and ‘music-based interventions are well-integrated’.

Differences were found within intervention groups regarding the influence of factors occurring for COMBI in relation to the facilitator’s ‘stable and well-functioning team’, with a median of 2.5 at baseline to 5.0 post-intervention. Within RCS, the median influence of four barriers decreased, indicating its influence was less than expected: ‘lack of support’ (median 3.0 to 1.0), ‘lack of motivation’, ‘scarce resources’, and ‘in general, implementing an intervention on the CHU takes a lot of time’ (medians from 4.0 to 2.0). For this last item, the median score changed similarly for CONTROL.

Between intervention groups at baseline, median differences were found for the facilitator ‘there are music activities in the care home’, with a median of 4.0 in COMBI and 2.0 in CONTROL. Post-intervention differences in influence of a factor were found for the barrier ‘participating in other projects’ with a median of 3.0 for GMT and 1.0 for COMBI; and also for the facilitator ‘project planning maintained’ with a median of 5.0 in COMBI and 3.0 in RCS. Respondents in COMBI experienced the largest influence of ‘keeping an intervention running is a lot of work’ (median 3.5), compared to the other groups.

## 4. Discussion

This process evaluation of a large multinational trial of group MBIs investigated expectations and experiences related to barriers and facilitators for the implementation of MBI for people with dementia living in care homes. Care home managers, care staff, and interventionists from Germany, the Netherlands, Norway, Türkiye, and the United Kingdom filled out a survey at baseline and after the six-month intervention period. Allocated intervention groups were group music therapy (GMT), recreational choir singing (RCS), the combination of both interventions (COMBI), or a control group receiving no structural MBIs (CONTROL). Key findings were organized and presented across three levels: the stakeholder perspective, the country perspective, and the intervention group perspective.

### 4.1. Interpretation of Main Findings

Group MBIs can successfully be implemented for people with dementia and depression in care home settings. Overall, respondents deemed the MBIs relevant and feasible, although not all stakeholders were certain what was expected of them during the project. The majority of respondents noticed an effect of the MBI on participants, although some were uncertain about the participants’ allocated intervention group. The principal barriers to the implementation of group MBIs in care homes for people with dementia include a lack of support, changes in employees, and a lack of motivation in those involved. Important facilitators to reinforce are: people involved in facilitating and stimulating implementation, stable and well-functioning teams, clear communication, and maintaining project planning. Contrary to expectations, the barriers keeping an MBI running are a lot of work, time needed to implement an intervention in general and an MBI specifically, high workload of care staff, lack of care staff, and discontinuity due to COVID were all frequent than respondents had anticipated. This finding contrasts with much of the literature that highlights staff burden as a key implementation barrier. One possible explanation is that the external contracting of interventionists in some countries (e.g., Germany, Türkiye) reduced the perceived workload for in-house staff. Alternatively, this could reflect a selection bias where more motivated or better-resourced care homes were included in the trial.

Low confidence in sustainability may reflect a lack of structural funding, institutional support for long-term integration, or insufficient training and handover mechanisms. Additionally, implementation may have been perceived as a project-based initiative rather than an embedded part of routine care. The majority of respondents were unaware of any plans for the sustainable adoption of the MBI after the trial. The uncertainty around sustainability highlights the need to consider long-term implementation outcomes, as described in Nilsen’s framework, where factors such as acceptability, appropriateness, and feasibility need to be aligned with institutional capacity for maintenance ([21]). Future work should consider strategies for scale and spread, which require not only local stakeholder engagement but also alignment with broader system-level levers such as national dementia strategies, professional training structures, and reimbursement pathways. Engaging key stakeholders is essential: care home managers are pivotal for policy making and sustainable adoption of music interventions, whereas commitment and involvement of care staff are needed for day-to-day implementation. Even though this study did not specifically focus on policy context and related financial and systemic barriers for implementation of music interventions for people with dementia living in care homes, its approach of differentiating perspectives of care home management, care staff and interventionists sheds light on how differences within the system are seen, depending on the respondent’s role in it.

#### 4.1.1. Country-Perspective

The contexts in which complex interventions such as these are implemented can have a large influence on intervention outcomes ([25]), although contextual factors and their interaction with interventions are often not taken into full consideration ([33]). However, the question is not only whether an intervention is effective, but also under what circumstances it can reach its full potential in daily practice. Possible contextual factors that may influence external validity include, among others, differences between cultures, countries, healthcare systems, selection of care homes and care home residents, intervention protocols, and clinically-relevant outcome measures ([23]). Indeed, several findings from this study may be due to cultural and healthcare system differences. For instance, German and Turkish respondents anticipated a high presence and influence of a ‘flat hierarchy’, whereas Dutch and Norwegian respondents anticipated a high presence and influence of implementation of the MBI taking a lot of time. The relatively low number of reported barriers in Germany may reflect a more centralized and externally managed implementation model, where interventionists were employed by the research institute rather than embedded in local care structures. This could reduce friction in implementation but may also limit local ownership or awareness of challenges. One difference in healthcare systems, for example, is that in Germany, no elderly care physicians and psychologists are employed by the care home, implying a less hierarchical structure ([7]). Moreover, in the Netherlands, interventionists were contracted by the care home organization, in Norway, they were employed by the municipalities, whereas in Germany and Türkiye, interventionists worked for the research institute conducting the MIDDEL trial. Those interventionists went to various CHUs to offer the assigned interventions to the participating residents. Another difference between countries was the level of education and certification of interventionists and their level of experience in applying music therapy in a clinical setting such as elderly care. To elaborate, in Türkiye, music therapy in general is not (yet) a recognized profession, whereas in the other countries it is. Hitherto in Türkiye, music therapy has been recognized as a technique which only trained medical doctors or dentists can apply, indicating that it is actually music in medicine which has been practiced ([29]). This suggests that, beyond cultural attitudes, structural elements like national certification systems and educational policy play a key role in shaping the scope and authority of professionals, which in turn may directly influence intervention uptake and sustainability. The high anticipated influence of ‘participating in other studies or projects’ in Turkish respondents could be explained by a difference in previous experiences with trials in care homes.

#### 4.1.2. Stakeholder-Perspective

Differences between stakeholders’ evaluation of the presence and influence of factors indicate that each stakeholder group held a different perspective, was involved in the trial in a different way, and therefore experienced different barriers and facilitators. Previous studies identified factors related to staff, communication, stimulation of implementation, and expectations to have a large influence on implementation ([8]; [10]). An ambassador or key contact person for MIDDEL was present at the CHU to facilitate implementation and maintain the project planning, which has previously demonstrated to be a valuable strategy to facilitate implementation ([8]). Implementation strategies between countries also differed, which could explain why Dutch respondents knew less about which tasks they had to fulfill for the MIDDEL trial. To elaborate, in the Netherlands, all care staff and managers—even if they were involved to a lesser extent—were asked to fill out the survey, whereas in the other countries, a selection of stakeholders was specifically asked to fill out the survey.

During the trial, care home managers had a key position in enhancing facilitators and minimizing barriers to the implementation of group MBIs at the care staff level. They are in the position to offer organizational support in the form of allocating time, education, and resources for care staff to facilitate and maintain implementation and maintenance of the MBIs ([27]). In terms of sustainability, intention and the effort of care home management are pivotal to the continuation and adoption of MBIs. [27] ([27]) also emphasized that involving care staff in protocol adaptations fostered a greater sense of ownership and responsibility for implementation. Similarly, stakeholders in the MIDDEL trial were unsure about their role and tasks during the trial. This may have hindered their involvement and sense of shared responsibility to maintain the MBIs.

#### 4.1.3. Intervention Group-Perspective

Differences observed in the anticipated presence of factors provide insight into the preconditions that differed between intervention groups prior to randomization to the allocated intervention group (GMT, RCS, COMBI, or CONTROL). In each group, the baseline percentage of respondents who expected factors to be present during the trial was much higher than the percentage of respondents who subsequently experienced factors to have been present. The effects that respondents noticed in care home residents were highest in RCS, followed by GMT and COMBI. An interesting next step would be to compare these findings with the results found in the MIDDEL effect study. Respondents in the CONTROL group also noticed an effect in residents, which could relate to half of the respondents being unaware of the exact intervention allocation. In addition, the personal attention provided by repeatedly interviewing residents individually in the control group may have had an influence.

The findings from this study highlight that the type and dose of MBI affect implementation. Stakeholders involved in the COMBI group encountered greater organizational challenges, as sessions had to be organized not twice but four times a week. Perceived barriers and facilitators varied depending on an individual’s role in the project. For care home managers initiating a new intervention, clear communication across organizational levels was particularly important, whereas for care staff, key facilitators included sufficient time, role clarity, and effective collaboration.

### 4.2. Strengths and Limitations

This study has several strengths. Its design and the questionnaire were based on previous studies into barriers and facilitators for implementation in nursing homes ([2]; [16]; [35]). The Leontjevas framework was selected over other established frameworks such as the Consolidated Framework for Implementation Research (CFIR) ([5]) and the RE-AIM (Reach, Effectiveness, Adoption, Implementation, Maintenance) framework ([12]). Leontjevas’ framework fitted the aim and scope of our process evaluation; was easy to embed in a multinational RCT; and has previously been applied in care home settings for people living with dementia. We opted for a repeated survey with closed questions from different stakeholders involved in the MIDDEL trial, to obtain insight into expectations and experiences related to the implementation of the trial and its interventions. A descriptive approach was chosen to analyze the large quantity of data. The survey was completed by four different groups of stakeholders, each with their own area of expertise, (work) experiences, and opinions, which in turn influenced their evaluation of the implementation. Each stakeholder group was involved in the project in a different way and could therefore experience factors (potential barriers and facilitators) differently. The questionnaire was completed at several time points, allowing us to gain insight into changes over time. It consisted solely of factual statements, enabling us to conduct the survey in multiple languages. This allowed us to make comparisons between countries with minimal language barriers.

This study also has several limitations. First, we drew on the Leontjevas model, which is widely used to evaluate psychosocial interventions in dementia care. However, we did not adopt a systematic theoretical or strategic approach to account for contextual factors that could help clarify what works, for whom, and under which circumstances—an important consideration highlighted in the review by Wyman ([33]). Furthermore, the process evaluation framework we used was not prescriptive, which limited the consistency and depth of its application. Second, strategies to interweave the survey in the MIDDEL trial differed by country. Before the start of the trial, it was decided that each country would try to approach 30% of the stakeholders for the process evaluation. Due to the size of the trial, it was not feasible for all participating countries to embed the process evaluation survey in the MIDDEL trial to the same extent. As a result, the sample size differs between countries and stakeholders. Each country could also decide for itself how to achieve this target number, with the result that in some countries, only stakeholders from certain units have been asked, and other units not at all. In addition, selection bias was likely: specific people were asked by MIDDEL researchers to share their perspective by completing the survey. Furthermore, the moment that the process evaluation was rolled out and completed differed across countries and participating homes, so that in one care home, the presence and impact of COVID (restrictions) may have been much more prominent than in homes that started the process evaluation at a later time. No information is available about those who did not fill in the survey. Respondents who completed the survey at baseline may differ from those who completed the survey post-intervention. These limitations may create an incomplete, biased picture of the overall implementation; therefore, findings should be interpreted with caution. One item in the survey that may have been ambiguous was the question about ‘any’ observed effect of the MBIs. One-third of respondents from the control group indicated that they noticed an effect in care home residents. The cause remains unknown: perhaps socially desired answers were given, a placebo effect occurred due to positive expectations related to participating in the trial, or respondents filled out the survey with other musical activities in mind, which were part of standard care in their care home. In addition, it is unknown whether the observed effects are valued in a positive or negative way.

### 4.3. Recommendations for Future Studies and Practice

Thorough documentation of process evaluation for complex interventions, like group MBIs, is essential to assess not only the intervention’s effectiveness, but also the effectiveness of the implementation strategies employed. A descriptive, theory-driven approach was chosen for data analysis in this study ([25]), although future studies focusing on process evaluation embedded in an RCT for psychosocial interventions might consider using a standardized questionnaire, where significance testing is also possible. Several survey items could be formulated differently in future trials. Specifically, in a future study, it could be valuable to differentiate between positive effects (such as joy) and negative effects (such as participant burden) when asking stakeholders whether they observed ‘any effect’. In future trials, a deliberate decision might be made whether or not to remind respondents at the start of the survey to which arm their CHU was allocated. On the one hand, this ensures that respondents complete the survey with the right intervention in mind. On the other hand, it is also valuable to know that respondents do not know exactly whether and which MBI the residents were offered. This study focused on the expectations and experiences of stakeholders. For a more complete picture of the implementation of the trial and the MBIs, it is advisable to assess multiple aspects of the process evaluation, such as intervention dose, session attendance, reliability, and validity of the MBIs by assessing treatment fidelity. For a complete picture of sustainable implementation, assessment of cost-effectiveness of the MBIs will be assessed.

Differences between countries were large, warranting further exploration of local/national barriers and facilitators for implementation. Evaluation of implementation could be explored in-depth for each participating country separately, as was carried out in the UK arm of the MIDDEL trial ([24]). The anecdotes described herein provide a vivid picture of the universal challenges that arise when implementing a group MBI in a care home setting. A next step in evaluating the implementation of group-based music interventions for people with dementia and depressive symptoms will be to connect main effect findings to the barriers to and facilitators for implementation, which were experienced, enabling translation of MBIs into day-to-day practice. A combination of elements of the RE-AIM framework and the CFIR framework could be appropriate to help systematically explain variation across contexts. In a follow-up paper, we intend to synthesize findings from the trial with findings from the process evaluation. This approach enables us to explore possible discrepancies between expectations and experiences and relate them to the measured effects of MBIs.

## 5. Conclusions

The primary objective of this process evaluation was, firstly, to gain insight into stakeholders’ expectations, which can be taken to reflect their satisfaction with a proposed interventional trial, its relevance, and feasibility. We also looked at their actual experience of the barriers to and facilitators for implementation of MBIs for people with dementia and depression in care homes. Findings of this process evaluation illustrate how implementing a complex intervention in care homes can be challenging, based on how potential barriers and facilitators, that is, organizational factors and factors related to the MBI (i.e., the type and dose), affect implementation and adoption of the MBI in the care home. Different MBIs come with different organizational and practical challenges for the organization, care home managers, interventionists, and care staff. When starting a randomized trial and implementing MBIs, consideration should be given to factors related to support from board/management, well-functioning teams, workload, and availability of care staff, clear communication, that people involved facilitate implementation, and that project planning is maintained. Findings of this process evaluation will be synthesized with the results of the MIDDEL trial to increase the sustainability and impact of MBIs in nursing homes.

## Figures and Tables

**Figure 1 behavsci-15-01004-f001:**
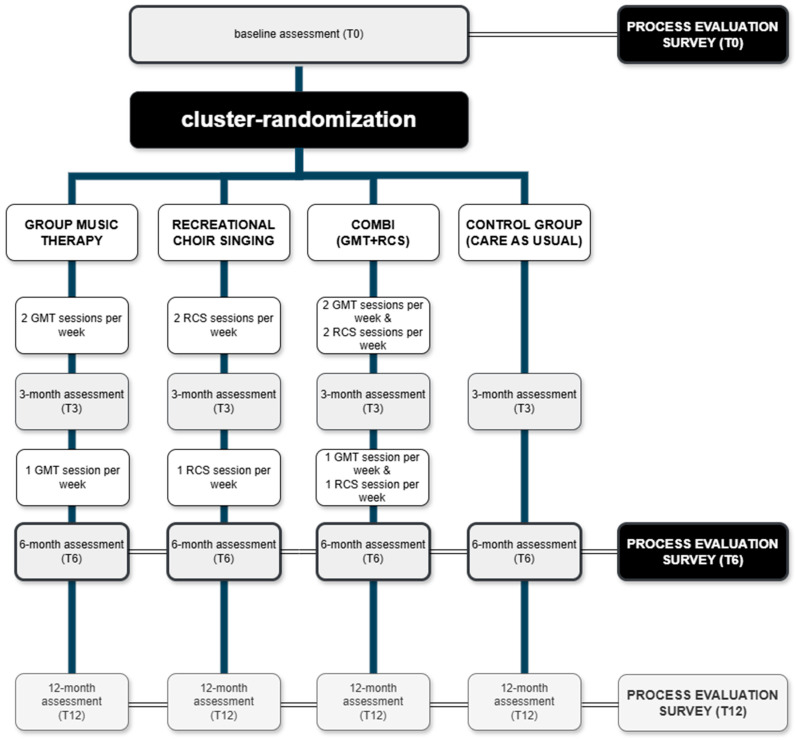
Flowchart illustrating the design of the process evaluation embedded in the main MIDDEL trial.

**Figure 2 behavsci-15-01004-f002:**
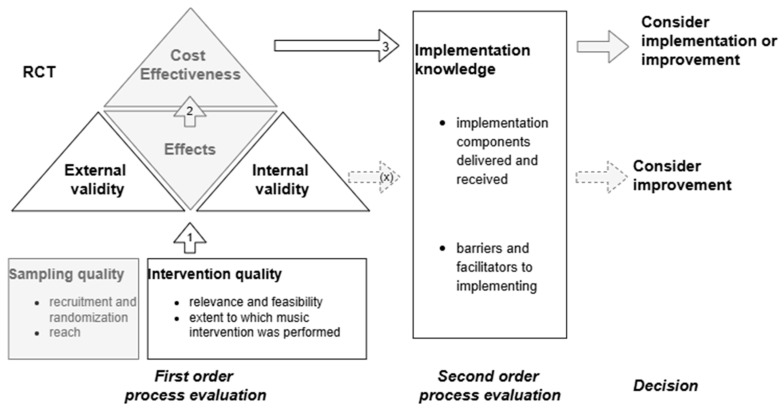
Path diagram illustrating the operationalization of elements derived from the framework ([16]) for process evaluation within the MIDDEL trial. Elements presented in gray fall outside the scope of the current study.

**Figure 3 behavsci-15-01004-f003:**
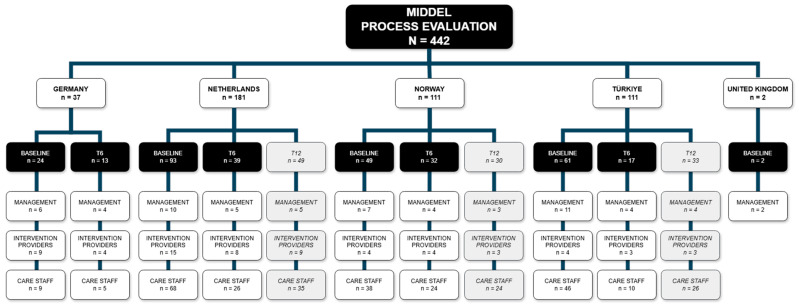
Overview of MIDDEL process evaluation surveys collected per country, presented per timepoint and stakeholder group.

**Figure 4 behavsci-15-01004-f004:**
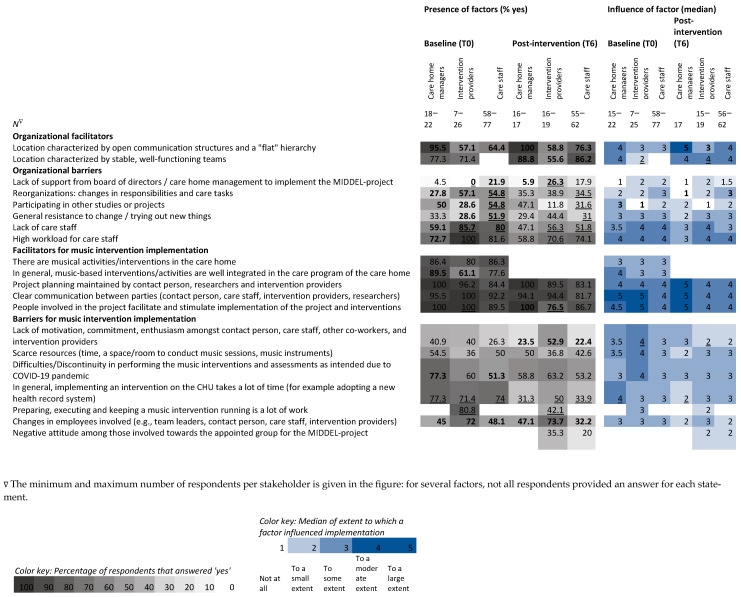
Heatmap visualizing the percentage of stakeholders that expected (T0) and experienced (T6) barriers and facilitators to be present during the MIDDEL trial and median scores of expected and experienced influence of those factors on the implementation of the MIDDEL trial and music interventions. Bold numbers indicate between-group differences, underscored numbers indicate within-group differences ≥ 20%/≥2.0 points on the median.

**Figure 5 behavsci-15-01004-f005:**
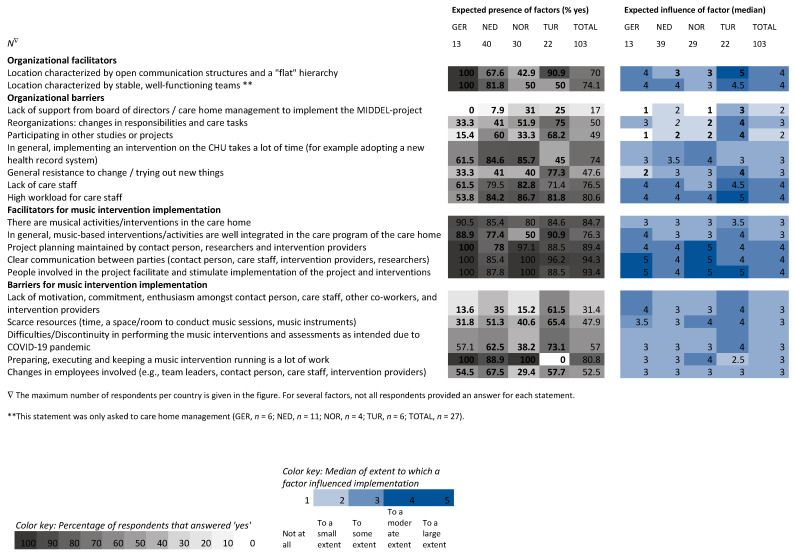
Heatmap visualizing the percentage of respondents per country that expected barriers and facilitators to be present during the MIDDEL trial and median scores of expected influence of those factors on the implementation of the MIDDEL trial and music interventions. Bold numbers indicate between-country differences ≥ 20%/≥2.0 points on the median.

**Figure 6 behavsci-15-01004-f006:**
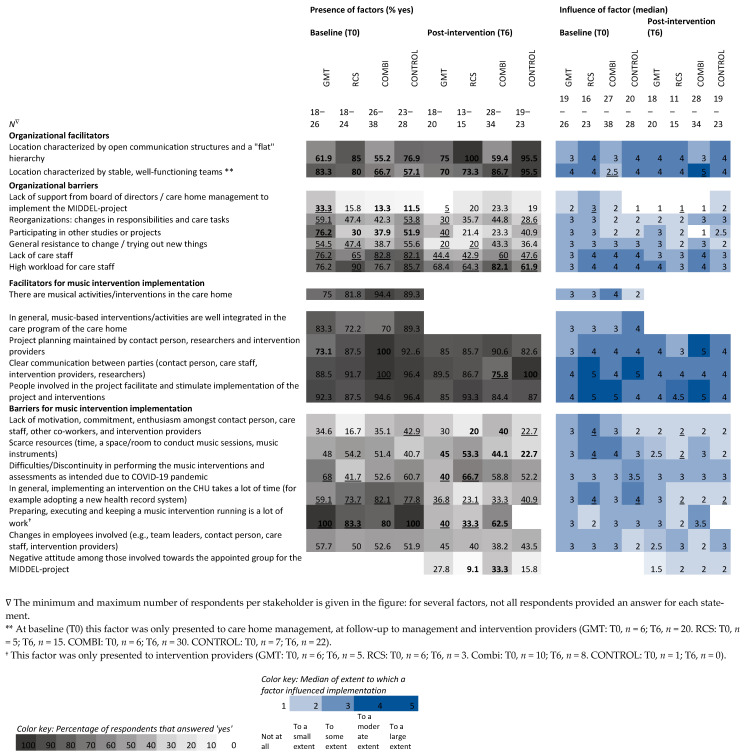
Heatmap visualizing the percentage of respondents in each intervention group who expected and experienced a hindering or facilitating factor to be present during the MIDDEL trial and median scores of expected and experienced influence of those factors on the implementation of the MIDDEL trial and music interventions. Bold numbers indicate between-group differences, underscored numbers indicate within-group differences ≥ 20%/≥2.0 points on the median.

**Table 1 behavsci-15-01004-t001:** Descriptive characteristics of stakeholder groups at baseline (T0) and post-intervention (T6).

Stakeholder Demographics per Timepoint	T0	T6
**Care home management (*n*)**	**22**	**17**
Age, *n*	18	16
18–29, *n* (%)	4 (22.2%)	3 (18.8%)
30–39, *n* (%)	5 (27.8%)	4 (25.0%)
40–49, *n* (%)	2 (11.1%)	4 (25.0%)
50–59, *n* (%)	5 (27.8%)	3 (18.8%)
over 60, *n* (%)	2 (11.1%)	2 (12.5%)
Sex, *n*	22	17
Female, *n* (%)	16 (72.7%)	15 (88.2%)
Work experience, *n*	22	16
Less than 2 years, *n* (%)	6 (27.3%)	7 (43.8%
2 to 5 years, *n* (%)	6 (27.3%)	2 (12.5%)
5 to 10 years, *n* (%)	3 (13.6%)	2 (12.5%)
More than 10 years, *n* (%)	7 (31.8%)	5 (31.3%)
Function, *n*	19	14
Location manager (department leader), *n* (%)	7 (36.8%)	4 (28.6%)
Staff officer, *n* (%)	1 (5.3%)	-
Supervisor (organization leader), *n* (%)	6 (31.6%)	6 (42.9%)
Other	5 (26.3%)	4 (28.6%)
Familiar with music interventions, *n*	20	
GMT, *n* (%)	2 (10.0%)	
RCS, *n* (%)	7 (35.0%)	
Both, *n* (%)	7 (35.0%)	
Neither, *n* (%)	4 (20.0%)	
Involved during the full trial length, *n*		15
Yes, *n* (%)		14 (93.3%)
Group allocation, n		14
Correct, *n* (%)		9 (64.3%)
Partially correct, *n* (%)		1 (7.1%)
Incorrect, *n* (%)		4 (28.6%)
Duration to start after baseline assessment, *n*		14
2 weeks or less, *n* (%)		2 (14.3%)
2–4 weeks, *n* (%)		4 (28.6%
I do not know, *n* (%)		7 (50.0%
Not applicable, *n* (%)		1 (7.1%)
Familiar with Ambassador/contact person, *n*	19	16
Yes, *n* (%)	17 (89.5%)	12 (75.0%)
I do not know, *n* (%)	2 (10.5%	4 (25.0%)
**Intervention providers (*n*)**	**26**	**19**
Age, *n*	26	17
18–29, *n* (%)	10 (38.5%)	7 (41.2%)
30–39, *n* (%)	4 (15.4%)	2 (11.8%)
40–49, *n* (%)	3 (11.5%)	2 (11.8%)
50–59, *n* (%)	7 (26.9%)	4 (23.5%)
over 60, *n* (%)	2 (7.7%)	2 (11.8%)
Sex, *n*	26	19
Female, *n* (%)	21 (80.8%)	15 (78.9%)
Work experience, *n*	25	15
Less than 2 years, *n* (%)	8 (32.0%)	5 (33.3%)
2 to 5 years, *n* (%)	11 (44.0%)	8 (53.3%)
5 to 10 years, *n* (%)	3 (12.0%)	2 (13.3%)
More than 10 years, *n* (%)	3 (12.0%)	-
Working at CHU before this trial, *n*	25	
Yes, *n* (%)	7 (28.0%)	
Involved during the full trial length, *n*		18
Yes, *n* (%)		16 (88.9%)
Group allocation, *n*		16
Correct, *n* (%)		12 (75.0%)
Partially correct, *n(*%)		1 (6.3%)
Incorrect, *n* (%)		3 (18.8%)
Duration to start after baseline assessment, *n*		19
2 weeks or less, *n* (%)		7 (36.8%)
2–4 weeks, *n* (%)		4 (21.1%)
6–8 weeks, *n* (%)		1 (5.3%)
I do not know, *n* (%)		7 (36.8%)
Familiar with Ambassador/contact person, yes n (%)		19 (100.0%)
**Care staff (*n*)**	**81**	**71**
Age, *n*	67	71
18–29, *n* (%)	13 (19.4%)	11 (20.8%)
30–39, *n* (%)	17 (25.4%)	10 (18.9%)
40–49, *n* (%)	9 (13.4%)	12 (22.6%)
50–59, *n* (%)	17 (25.4%)	15 (28.3%)
over 60, *n* (%)	11 (16.4%)	5 (9.4%)
Sex, *n*	81	63
Female, *n* (%)	71 (87.7%)	56 (88.9%)
Work experience, *n*	71	43
Less than 2 years, *n* (%)	2 (2.8%)	-
2 to 5 years, *n* (%)	9 (12.7%)	3 (7.0%)
5 to 10 years, *n* (%)	16 (22.5%)	10 (23.3%)
More than 10 years, *n* (%)	44 (62.0%)	30 (69.8%)
Function, *n*	65	52
Registered nurse, *n* (%)	15 (23.1%)	12 (23.1%
Enrolled nurse, *n* (%)	34 (52.3%)	33 (63.5%)
Personal care attendant, *n* (%)	6 (9.2%)	5 (9.6%)
Allied health professional, *n* (%)	5 (7.7%)	-
Leisure staff, *n* (%)	1 (1.5%)	2 (3.8%)
Other, *n* (%)	4 (6.2%)	-
Familiar with music interventions, *n*	81	
GMT, *n* (%)	16 (19,8%)	
RCS, *n* (%)	22 (27.2%)	
Both, *n* (%)	27 (33.3%)	
Neither, *n (*%)	16 (19.8%)	
Involved during the full trial length, *n*		58
Yes, *n* (%)		50 (86.2%)
Group allocation, *n*		65
Correct, *n* (%)		32 (49.2%)
Partially correct, *n* (%)		4 (6.2%)
Incorrect, *n* (%)		29 (44.6%)
Duration to start after baseline assessment, *n*		59
2 weeks or less, *n* (%)		5 (8.5%)
2–4 weeks, *n* (%)		5 (8.5%)
4–6 weeks, *n* (%)		2 (3.4%)
6–8 weeks, *n* (%)		5 (8.5%)
I do not know, *n* (%)		41 (69.5%)
Not applicable, *n* (%)		1 (1.7%)
Familiar with Ambassador/contact person, *n*	78	59
Yes, *n* (%)	49 (62.8%)	45 (76.3%)
No, *n* (%)	5 (6.4%)	2 (3.4%)
I do not know, *n* (%)	24 (30.8%)	12 (20.3%)
Group allocation across stakeholders, *n*		95
Correct, *n* (%)		53 (55.8%)
Partially correct, *n* (%)		6 (6.3%)
Incorrect, *n* (%)		36 (37.9%)

**Table 2 behavsci-15-01004-t002:** Stakeholders’ expectations (T0) and experiences (T6) related to relevance, feasibility, and implementation of the trial and music interventions.

Expectations at Baseline (T0)		Care Home Managers (*n* = 23)	Intervention Providers (*n* = 25)	Care Staff (*n* = 80)
**Relevance and feasibility of the organization** (level of agreement *)				
Participating in the project will be relevant (meaningful, fitting, important) for this care home organization	*n*	23		
	Median [IQR]	4.0 [4.0–5.0]		
Recreational choir singing will be relevant for the care home residents with dementia and depressive symptoms	*n*	23	25	80
	Median [IQR]	5.0 [4.0–5.0]	5.0 [4.0–5.0]	5.0 [4.0–5.0]
Group music therapy will be relevant for the care home residents with dementia and depressive symptoms	*n*	23	25	78
	Median [IQR]	5.0 [4.0–5.0]	5.0 [4.0–5.0]	4.0 [4.0–5.0]
Recreational choir singing will fit well into the day-to-day practice of the participating CHU(s)	*n*	23	25	78
	Median [IQR]	4.0 [4.0–5.0]	4.0 [3.0–5.0]	4.0 [4.0–5.0]
Group music therapy will fit well into the day-to-day practice of the participating CHU(s)	*n*	23	25	79
	Median [IQR]	4.0 [4.0–5.0]	4.0 [3.5–5.0]	4.0 [3.0–5.0]
I know which tasks I have to fulfill for the MIDDEL project	*n*	22	26	76
	Median [IQR]	4.0 [3.0–5.0]	**5.0 [4.0–5.0]**	**3.0 [2.0–4.0]**
**Implementation expectations at baseline (T0)** (to what extent ∗)				
Anticipated facilitators will be reinforced	*n*	20	21	77
	Median [IQR]	3.0 [3.0–4.0]	4.0 [3.5–4.5]	3.0 [3.0–4.0]
Anticipated barriers will be resolved	*n*	20	17	77
	Median [IQR]	4.0 [3.3–4.0]	4.0 [3.5–5.0]	4.0 [3.0–4.0]
**Experiences post-intervention (T6)**		**Care home managers** **(*n* = 17)**	**Intervention providers** **(*n* = 19)**	**Care staff** **(*n* = 65)**
**Relevance and feasibility of the organization** (level of agreement *)				
Participating in the project was relevant (meaningful, fitting, important) for this care home organization	*n*	16		
	Median [IQR]	4.5 [4.0–5.0]		
The music intervention was relevant for care home residents with dementia and depressive symptoms	*n*	13	17	54
	Median [IQR]	5.0 [4.0–5.0]	5.0 [4.0–5.0]	4.0 [3.0–5.0]
The music intervention was relevant to reduce depressive symptoms	*n*	13	17	52
	Median [IQR]	4.0 [3.5–5.0]	4.0 [4.0–5.0]	3.0 [3.0–4.0]
The music intervention fit well into the day-to-day practice of the participating CHU(s)	*n*	14	17	56
	Median [IQR]	4.5 [4.0–5.0]	4.0 [3.0–5.0]	3.0 [2.0–3.0]
The music intervention turned out to be too complex to use	*n*	15	19	50
	Median [IQR]	2.0 [2.0–3.0]	3.0 [1.0–4.0]	3.0 [2.0–3.0]
The music intervention was in line with how we are used to working	*n*	13	18	54
	Median [IQR]	4.0 [2.5–4.5]	4.0 [3.0–5.0]	3.5 [3.0–4.0]
I would recommend the music intervention to other care homes	*n*	14	17	57
	Median [IQR]	5.0 [3.8–5.0]	5.0 [4.5–5.0]	4.0 [4.0–5.0]
**Effect**				
Did you notice any effect of the music intervention(s)?	*n*	17	19	65
	Yes, on the CHU, *n* (%)	5 (29.4%)	7 (36.8%)	8 (12.3%)
	Yes, in care staff, *n* (%)	6 (35.3%)	9 (47.4%)	7 (10.8%)
	Yes, in residents, *n* (%)	11 (64.7%)	18 (94.7%)	28 (43.1%)
	No, *n* (%)	0 (0.0%)	0 (0.0%)	16 (24.6%)
	I do not know, *n* (%)	2 (11.8%)	2 (10.5%)	10 (15.4%)
	Not applicable, *n* (%)	3 (17.6%)	0 (0.0%)	8 (12.3%)
**Satisfaction** (level of satisfaction ⌑)				
To what extent were you satisfied with the implementation of the MIDDEL-project?	*n*	17	17	59
	Median [IQR]	5.0 [3.5–5.0]	4.0 [3.0–4.5]	4.0 [3.0–4.0]
**Implementation** (to what extent ∗)				
To what degree was the music intervention implemented on the CHU?	*n*	17	18	57
	Median [IQR]	4.0 [4.0–5.0]	4.0 [3.8–5.0]	4.0 [3.0–5.0]
Anticipated facilitators will be reinforced	*n*	17	19	58
	Median [IQR]	3.0 [2.5–4.0]	3.0 [3.0–4.0]	3.0 [2.0–4.0]
Anticipated barriers will be resolved	*n*	17	17	59
	Median [IQR]	4.0 [3.0–4.0]	4.0 [3.0–4.0]	3.0 [2.0–4.0]
**Sustainability**				
Will the music intervention continue on the participating CHU(s) after the 6-month intervention period?	*n*	16	19	62
	Yes, *n* (%)	3 (18.8%)	1 (5.3%)	6 (9.7%)
	I do not know, *n* (%)	11 (68.8%)	11 (57.9%)	50 (80.6%)

* Answer options: 1 = Strongly disagree, 2 = Disagree, 3 = Neither agree nor disagree, 4 = Agree, 5 = Strongly agree. ∗ Answer options: 1 = Not at all, 2 = To a small extent, 3 = To some extent, 4 = To a moderate extent, 5 = To a large extent. ⌑ Answer options: 1 = Not satisfied, 2 = Hardly satisfied, 3 = Somewhat satisfied, 4 = Satisfied, 5 = Completely satisfied.

**Table 3 behavsci-15-01004-t003:** Countries’ median (IQR) baseline expectations (T0) related to relevance, feasibility, and implementation of the trial and music interventions.

		Germany (*n* = 24)	The Netherlands (*n* = 93)	Norway (*n* = 49)	Türkiye (*n* = 61)	Total ** (*n* = 297)
**Relevance and feasibility of the organization** (level of agreement *), median [IQR]						
Participating in the project will be relevant (meaningful, fitting, important) for this care home organization ∇	*n*	6	5	4	6	21
	Median [IQR]	4.0 [3.0–5.0]	4.0 [4.0–4.5]	4.5 [3.3–5.0]	4.0 [3.8–5.0]	4.0 [4.0–5.0]
Recreational choir singing will be relevant for the care home residents with dementia and depressive symptoms	*n*	21	43	36	26	126
	Median [IQR]	5.0 [4.0–5.0]	4.0 [4.0–5.0]	5.0 [4.3–5.0]	4.0 [3.8–5.0]	5.0 [4.0–5.0]
Group music therapy will be relevant for the care home residents with dementia and depressive symptoms	*n*	21	43	35	25	124
	Median [IQR]	4.0 [4.0–5.0]	4.0 [4.0–5.0]	5.0 [4.0–5.0]	4.0 [4.0–5.0]	4.5 [4.0–5.0]
Recreational choir singing will fit well into the day-to-day practice of the participating CHU(s)	*n*	21	43	34	26	124
	Median [IQR]	4.0 [3.0–4.5]	4.0 [4.0–5.0]	5.0 [4.0–5.0]	4.0 [3.8–5.0]	4.0 [4.0–5.0]
Group music therapy will fit well into the day-to-day practice of the participating CHU(s)	*n*	21	43	35	26	125
	Median [IQR]	4.0 [3.0–4.0]	4.0 [4.0–5.0]	5.0 [4.0–5.0]	4.0 [3.0–5.0]	4.0 [4.0–5.0]
I know which tasks I have to fulfill for the MIDDEL project	*n*	21	43	32	26	122
	Median [IQR]	**5.0 [4.0–5.0]**	**3.0 [2.0–4.0]**	4.0 [1.3–5.0]	4.0 [3.0–4.0]	4.0 [3.0–5.0]
**Implementation**, median [IQR]						
Anticipated facilitators will be reinforced	*n*	19	40	34	25	118
	Median [IQR]	4.0 [3.0–4.0]	3.0 [3.0–4.0]	4.0 [3.0–4.0]	4.0 [1.5–4.5]	4.0 [3.0–4.0]
Anticipated barriers will be resolved	*n*	16	39	34	25	114
	Median [IQR]	4.0 [3.0–4.0]	4.0 [3.0–4.0]	4.0 [3.0–4.0]	4.0 [2.5–4.5]	4.0 [3.0–4.0]

* Answer options: 1 = Strongly disagree, 2 = Disagree, 3 = Neither agree nor disagree, 4 = Agree, 5 = Strongly agree. ** As there were only two surveys from the United Kingdom, these surveys are excluded from this analysis. ∇ This statement was only applicable for care home management.

**Table 4 behavsci-15-01004-t004:** Expectations (T0) and experiences (T6) across intervention groups related to relevance, feasibility, and implementation of the trial and music interventions.

Expectations at Baseline (T0)		GMT (*n* = 26)	RCS (*n* = 24)	COMBI (*n* = 39)	CONTROL (*n* = 30)
**Relevance and feasibility** (level of agreement *)					
Participating in the project will be relevant (meaningful, fitting, important) for this care home organization	*n*	4	3	4	6
	Median [IQR]	**4.5 [4.0–5.0]**	**3.0 [3.0]**	4.0 [4.0–4.8]	4.5 [3.0–5.0]
Recreational choir singing will be relevant for the care home residents with dementia and depressive symptoms	*n*	25	24	39	30
	Median [IQR]	4.0 [3.5–5.0]	5.0 [4.0–5.0]	5.0 [4.0–5.0]	4.0 [4.0–5.0]
Group music therapy will be relevant for the care home residents with dementia and depressive symptoms	*n*	26	22	39	29
	Median [IQR]	4.0 [4.0–5.0]	5.0 [4.0–5.0]	5.0 [4.0–5.0]	4.0 [4.0–5.0]
Recreational choir singing will fit well into the day-to-day practice of the participating CHU(s)	*n*	24	24	39	29
	Median [IQR]	4.0 [3.0–5.0]	4.0 [4.0–5.0]	4.0 [4.0–5.0]	4.0 [3.0–5.0]
Group music therapy will fit well into the day-to-day practice of the participating CHU(s)	*n*	26	23	39	29
	Median [IQR]	4.0 [3.0–5.0]	4.0 [3.0–5.0]	4.0 [3.0–5.0]	4.0 [4.0–5.0]
I know which tasks I have to fulfill for the MIDDEL project	*n*	25	23	37	29
	Median [IQR]	3.0 [1.5–5.0]	4.0 [3.0–4.0]	4.0 [3.0–5.0]	3.0 [2.0–4.0]
**Implementation**					
Anticipated facilitators will be reinforced	*n*	25	22	37	27
	Median [IQR]	3.0 [3.0–4.0]	4.0 [3.8–4.3]	4.0 [3.0–5.0]	3.0 [3.0–4.0]
Anticipated barriers will be resolved	*n*	23	22	36	26
	Median [IQR]	4.0 [3.0–4.0]	4.0 [4.0–4.0]	4.0 [3.0–4.0]	4.0 [3.0–4.0]
**Experiences at post-intervention (T6)**		**GMT** **(*n* = 23)**	**RCS** **(*n* = 17)**	**COMBI** **(*n* = 35)**	**CONTROL** **(*n* = 25)**
**General**					
There was an Ambassador/contact person for the MIDDEL project.	*n*	19	15	31	23
	Yes, *n* (%)	13 (68.4%)	12 (80.0%)	28 (90.3%)	17 (73.9%)
	I do not know, *n* (%)	5 (26.3%)	3 (20.0%)	3 (9.7%)	5 (21.7%)
**Supervision**					
The supervision and guidance for the MIDDEL project were sufficient	*n*	4	3	7	
	Median [IQR]	4.5 [4.0–5.0]	5.0 [5.0–5.0]	5.0 [4.0–5.0]	
The supervision sessions throughout the MIDDEL project were useful	*n*	4	2	6	
	Median [IQR]	4.5 [4.0–5.0]	5.0 [5.0–5.0]	5.0 [4.0–5.0]	
**Instruction**					
Did you receive information explaining the MIDDEL-project?	*n*	11	7	19	18
	Yes, *n* (%)	6 (54.5%)	6 (85.7%)	14 (73.7%)	14 (77.8%)
	I do not know, *n* (%)	1 (9.1%)	1 (14.3%)	1 (5.3%)	2 (11.1%)
I knew which tasks I had to fulfill for the MIDDEL project	*n*	10	7	20	14
	Median [IQR]	3.0 [1.0–4.3]	4.0 [2.0–5.0]	4.0 [4.0–5.0]	4.0 [3.8–5.0]
I was satisfied with the information offered prior to the start of the project	*n*	12	8	20	16
	Median [IQR]	4.0 [1.3–5.0]	4.0 [3.0–4.8]	4.0 [3.0–5.0]	4.0 [3.0–5.0]
The information, instruction, and guidance for the MIDDEL project were sufficient	*n*	10	8	21	15
	Median [IQR]	3.5 [1.0–4.3]	4.0 [3.0–4.0]	4.0 [2.5–5.0]	4.0 [4.0–5.0]
It was relevant to receive information prior to the start of the MIDDEL project	*n*	11	7	23	15
	Median [IQR]	4.0 [3.0–5.0]	4.0 [3.0–4.0]	5.0 [3.0–5.0]	4.0 [4.0–5.0]
**Relevance and feasibility** (level of agreement *)					
Participating in the project was relevant (meaningful, fitting, important) for this care home organization	*n*	2	3	3	3
	Median [IQR]	**5.0 [5.0]**	4.0 [4.0]	4.0 [3.0]	**3.0 [3.0]**
The music intervention was relevant for care home residents with dementia and depressive symptoms *	*n*	23	17	32	15
	Median [IQR]	4.0 [3.0–5.0]	4.0 [3.5–5.0]	4.0 [3.0–5.0]	4.0 [4.0–5.0]
The music intervention was relevant to reduce depressive symptoms	*n*	22	17	32	14
	Median [IQR]	3.0 [3.0–5.0]	4.0 [3.0–5.0]	3.0 [3.0–4.0]	4.0 [3.0–5.0]
The music intervention fit well into the day-to-day practice of the participating CHU(s)	*n*	18	15	32	17
	Median [IQR]	5.0 [4.0–5.0]	4.0 [4.0–5.0]	4.0 [3.0–4.0]	5.0 [3.5–5.0]
The music intervention turned out to be too complex to use	*n*	17	15	32	14
	Median [IQR]	3.0 [1.5–3.0]	2.0 [1.0–2.0]	3.0 [2.0–4.0]	2.0 [1.0–3.0]
The music intervention was in line with how we are used to working	*n*	19	13	31	16
	Median [IQR]	3.0 [2.0–4.0]	4.0 [3.0–5.0]	3.0 [2.0–4.0]	4.5 [3.0–5.0]
I would recommend the music intervention to other care homes	*n*	19	15	31	17
	Median [IQR]	5.0 [4.0–5.0]	5.0 [4.0–5.0]	4.0 [3.0–5.0]	5.0 [4.0–5.0]
**Effect**					
Did you notice any effect of the music intervention(s)?	*n*	20	15	35	25
	Yes, on the CHU, *n* (%)	6 (30.0%)	3 (20.0%)	6 (17.1%)	3 (12.0%)
	Yes, in care staff, *n* (%)	4 (20.0%)	5 (33.3%)	9 (25.7%)	2 (8.0%)
	Yes, in residents, *n* (%)	12 (60.0%)	14 (93.3%)	20 (57.1%)	7 (28.0%)
	No, *n* (%)	2 (10.0%)		11 (31.4%)	3 (12.0%)
	I do not know, *n* (%)	5 (25.0%)	1 (6.7%)	2 (5.7%)	5 (20.0%)
	Not applicable, *n* (%)	3 (15.0%)			7 (8.0%)
**Satisfaction** (level of satisfaction ⌑)					
To what extent were you satisfied with the implementation of the MIDDEL-project?	*n*	19	15	32	21
	Median [IQR]	4.0 [3.0–4.0]	4.0 [4.0–5.0]	4.0 [3.0–4.8]	4.0 [3.0–4.5]
**Implementation** (to what extent ∗)					
To what degree was the music intervention implemented on the CHU?	*n*	19	15	31	22
	Median [IQR]	4.0 [3.0–4.0]	4.0 [3.0–4.0]	3.0 [2.0–4.0]	3.0 [2.0–4.0]
Anticipated facilitators were reinforced	*n*	20	15	32	22
	Median [IQR]	3.0 [3.0–4.0]	4.0 [3.0–4.0]	3.0 [2.0–4.0]	4.0 [3.0–4.0]
Anticipated barriers were resolved	*n*	20	15	32	22
	Median [IQR]	4.0 [3.0–4.0]	4.0 [4.0–5.0]	4.0 [3.0–5.0]	4.0 [2.3–5.0]
**Sustainability**					
Will the music intervention continue on the participating CHU(s) after the 6-month intervention period?	*n*	20	15	34	23
	Yes, *n* (%)	2 (10.0%)	2 (13.3%)	2 (5.9%)	4 (17.4%)
	I do not know, *n* (%)	15 (75.0%)	11 (73.3%)	27 (79.4%)	15 (65.2%)

* Answer options: 1 = Strongly disagree, 2 = Disagree, 3 = Neither agree nor disagree, 4 = Agree, 5 = Strongly agree. ∗ Answer options: 1 = Not at all, 2 = To a small extent, 3 = To some extent, 4 = To a moderate extent, 5 = To a large extent. ⌑ Answer options: 1 = Not satisfied, 2 = Hardly satisfied, 3 = Somewhat satisfied, 4 = Satisfied, 5 = Completely satisfied.

## Data Availability

Data can be obtained from the corresponding author upon request.

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
