# Peer review of "Barriers and Facilitators for Implementing Music Interventions in Care Homes for People with Dementia and Depression: Process Evaluation Results of the Multinational Cluster-Randomized MIDDEL Trial"

_behavsci, 2025, doi:10.3390/bs15081004_

Round 1
Reviewer 1 Report
Comments and Suggestions for Authors
I have attached a file.

Author Response
This manuscript offers a significant contribution by evaluating implementation processes in a large-scale, MBIs trial in elderly care homes. The authors effectively utilize Leontjevas’ framework for first- and second-order process evaluation, providing an important lens for understanding contextual factors that impact the adoption of MBIs. The use of multi-country data and stakeholder differentiation (managers, staff, interventionists) is a key strength. However, the manuscript would benefit from more robust theoretical integration, refinement in data presentation, and deeper analysis of stakeholder and country-level differences to maximize the translational value of the findings.
- We thank the reviewer for the positive evaluation of the relevance of our process evaluation. In our response below we addressed the key issues that were highlighted by the reviewer.
Theoretical Framing and Analytical Depth: While the use of Leontjevas’ framework is appropriate, it is largely descriptive. The manuscript would benefit from deeper theoretical engagement.
- We agree with the reviewer and added this as a limitation to the discussion. It now says: “We drew on the Leontjevas model, which is widely used to evaluate psychosocial interventions in dementia care. However, we did not adopt a systematic theoretical or strategic approach to account for contextual factors that could help clarify what works, for whom, and under which circumstances—an important consideration highlighted in the review by Wyman (Wyman et al., 2022). Furthermore, the process evaluation framework we used was not prescriptive, which limited the consistency and depth of its application.” (line 613).
Clarity and Language: While the manuscript is generally well-written, certain sections (e.g., descriptions of stakeholder differences) are overly wordy and should be edited for clarity and conciseness. Avoid redundant phrases such as "participating in other projects" and "lack of support" being repeatedly listed without further elaboration.
- We aimed to rewrite the manuscript aiming for less wordiness, with increased clarity and conciseness.
Methods: The Methods section presents a comprehensive and well-structured description of the process evaluation embedded in the MIDDEL trial. The use of Leontjevas' framework to distinguish between first- and second-order process data is methodologically sound and allows for a nuanced understanding of both implementation quality and influencing contextual factors. The survey development process, including translation and localization, is described in sufficient detail and demonstrates thoughtful cross-national coordination. However, the section is at times overly dense. The description of the survey instrument could benefit from more concise summarization, particularly regarding minor stakeholder-specific wording differences, which could be moved to supplementary materials.
- We agree with the reviewer that the section describing survey development is detailed. As suggested by the reviewer, we summarized this description in the manuscript and moved elaborations on minor stakeholder-specific differences to supplementary materials. The following section was moved to Supplementary file 1: “To illustrate, the baseline (T0) survey questions and statements were formulated in simple future tense (e.g., “Recreational choir singing will be relevant for the care home residents with dementia and depressive symptoms.” and “Please indicate below for each factor (regardless of whether it is hindering or facilitating) to what extent you expect its presence may influence the implementation of the intervention.”). Items of the post-intervention survey T6 were formulated in the past tense (e.g., “To what extent were you satisfied with the implementation of the MIDDEL-project?” and “Please indicate below for each factor or statement whether this was applicable within your organization during the MIDDEL-project”). A single statement, for which stakeholders indicated to what extent a factor was present, was worded differently per stakeholder. The statement for managers was “Lack of support from board of directors to implement the MIDDEL-project”, whereas the statement for care staff was “Lack of support from care home management to implement the MIDDEL project.”
Results: The results are reported in an overly descriptive manner. The manuscript would benefit from deeper interpretation of the implications of divergent stakeholder experiences, particularly regarding the difference between expected and actual barriers (e.g., “lack of support” or “motivation”). The discussion would benefit from a closer integration with established implementation science frameworks. Incorporating constructs from models such as the Consolidated Framework for Implementation Research (CFIR) or Normalization Process Theory (NPT) could provide a more robust and theoretically grounded interpretation of the findings.
- Thank you for this suggestion. We consulted several papers that incorporated established the abovementioned established frameworks and agree they will be appropriate to help explain variation in adoption, fidelity and sustainability across contexts. We intend to provide this deeper interpretation of implications in a follow-up paper, in which we synthesize process evaluation findings with findings from the effect paper of the main MIDDEL trial, to gain deeper understanding of the role and effect of barriers and facilitators regarding implementation of music-based interventions. We described our motivation to use the Leontjevas framework for the design of the process evaluation in the revised manuscript. In the Discussion section we now added: “The Leontjevas framework was selected over other established frameworks such as the Consolidated Framework for Implementation Research (CFIR) (Damschroder 2022) and the RE-AIM (Reach, Effectiveness, Adoption, Implementation, Maintenance) framework (Holtrop 2021). Leontjevas’ framework fit the aim and scope of our study; is easy to embed in a multinational RCT; and has commonly been applied in care home settings for people living with dementia. A next step in evaluating the implementation of group-based music interventions will be to connect main effect findings with barriers and facilitators for implementation, enabling a translation to day-to-day practice. A combination of elements of the RE-AIM framework and CFIR framework could be appropriate to help systematically explain variation across contexts” (line 591). We would like to hear if this point requires further attention.
Additionally, the use of underlining or bold text in the main body of the manuscript is unnecessary and should be avoided.
- Thank you for addressing this. The purpose of using bold and underlining in the main text of the manuscript was to help the reader differentiate between 'between' and 'within' groups. Since it is not considered helpful for the reading experience, we removed the bold and italics from the main body of text.
Discussion: The discussion touches on contextual differences across countries and stakeholders but lacks a synthesized interpretation of why these differences exist or how they interact with systemic structures like healthcare policy, organizational culture, or professional roles. The discussion currently lacks sufficient depth in analyzing country-level and group-level differences. Why, for instance, did Germany report fewer barriers, and what does this say about system-level factors or cultural context? The finding that barriers related to care staff workload were "less than expected" warrants more critical exploration, particularly since staff burden is a well-documented barrier in care home settings.
- We thank the reviewer for pointing this out. As suggested, we added a synthesized interpretation of why these differences exist and how they interact with systemic structures. We added in paragraph 4.1 Interpretation of main findings: “This finding contrasts with much of the literature that highlights staff burden as a key implementation barrier. One possible explanation is that the external contracting of interventionists in some countries (e.g., Germany, Türkiye) reduced the perceived workload for in-house staff. Alternatively, this could reflect a selection bias where more motivated or better-resourced care homes were included in the trial” (line 490).
- In paragraph 4.1.1 Country perspective, we added: “The relatively low number of reported barriers in Germany may reflect a more centralized and externally managed implementation model, where interventionists were employed by the research institute rather than embedded in local care structures. This could reduce friction in implementation but may also limit local ownership or awareness of challenges.” And “This suggests that beyond cultural attitudes, structural elements like national certification systems and educational policy play a key role in shaping the scope and authority of professionals, which may directly influence intervention uptake and sustainability” (line 526).
- In our experience, it will also be valuable to use findings of the main effect study to give direction to which level warrants further in-depth analysis. This will include synthesis with main findings. We clarified these intentions in the manuscript, so that the descriptive nature of the current manuscript makes sense as a first step for translation of these findings to day-to-day practice. We added the following to the end of the discussion: “A next step in evaluating the implementation of group-based music interventions for people with dementia and depressive symptoms will be to connect main effect findings with experienced barriers and facilitators for implementation, enabling a translation to day-to-day practice. A combination of elements of the RE-AIM framework and CFIR framework could be appropriate to help systematically explain variation across contexts. In a follow-up paper, we intend to synthesize findings of the effect study with findings of the process evaluation. This approach enables us to explore possible discrepancies be-tween expectations and experiences and subsequently relate them to findings of the effect study” (line 668).
Clarify why perceived benefits were reported even in the control group, as this may suggest contamination or misattribution.
- We agree this issue could be elaborated on further. The authors added the following: “Respondents in the CONTROL group also noticed an effect in residents, which could relate to half of the respondents being unaware of the exact intervention allocation. In addition, the personal attention provided by repeatedly interviewing residents individually in the control group may have had an influence” (line 580).
Conclusions: The conclusion effectively summarizes key findings from the process evaluation and highlights important contextual and organizational factors affecting the implementation of MBIs. However, it could be strengthened more clearly by more clearly restating the study’s primary aim and explicitly linking how these process insights inform or qualify the interpretation of the MIDDEL trial’s outcomes.
- Following the reviewer’s suggestion, we restated the study’s primary aim and explicitly linked this to how we intend to synthesize it with main findings of the MIDDEL trial. We described how process evaluation findings will inform the interpretation of these main outcomes. We started the conclusion with: “The primary objective of this process evaluation was to gain insight into satisfaction, relevance and feasibility with; and barriers and facilitators for implementation of MBIs for people with dementia and depression in care homes, from the perspective of care home managers, care staff, and care home residents with dementia and depressive symptoms.” The final sentence of the conclusion has been adjusted to: “Findings of this process evaluation will be synthesized with results of primary outcomes of the MIDDEL trial, to increase sustainability and impact of MBIs in nursing homes” (line 678).
Reviewer 2 Report
Comments and Suggestions for Authors
Thank you for submitting this highly interesting manuscript. The manuscript reports on the process evaluation component of the large-scale, multinational MIDDEL trial, which studied the implementation of music-based interventions (MBIs) for residents in care homes with dementia and depressive symptoms. The study aims to identify barriers and facilitators to MBI implementation by analysing survey responses from stakeholders across five countries. It contributes meaningfully to the literature on complex interventions in geriatric care and music therapy by foregrounding process-related insights in a robust and methodologically transparent manner.
The topic is of high relevance given the global rise in dementia prevalence and the increasing attention to non-pharmacological interventions for neuropsychiatric symptoms in the elderly. The focus on music interventions, which are low-cost and socially engaging, aligns with current trends in person-centred care.
The study design is methodologically robust. Embedding the process evaluation within a cluster-randomised trial and following the framework of Leontjevas et al. ensures theoretical grounding and systematic data collection. The sample size is considerable and covers multiple stakeholder perspectives.
Including data from five countries enhances the study’s generalisability and allows for valuable cross-cultural comparisons in implementation practices and experiences. The dual use of first-order and second-order process data is well-conceived. The integration of quantitative Likert-scale data and binary responses provides a balanced descriptive picture. The inclusion of heatmaps to illustrate intergroup and intragroup differences is highly effective and increases the manuscript’s communicative clarity.
There are 5 issues that need extra attention:
Limited Depth in Interpretative Analysis While the manuscript offers a thorough descriptive overview of stakeholder responses regarding barriers and facilitators, it falls short in exploring why these differences occurred or what they imply for practice and theory. For example, it is reported that certain barriers—such as "high workload" and "lack of motivation"—were anticipated but experienced to a lesser degree than expected. However, the manuscript does not sufficiently unpack potential reasons behind this discrepancy.
Recommendation: The authors could draw on literature about adaptive capacity in care settings or the motivational dynamics of care staff in relation to creative interventions. They might consider whether familiarity with music-based activities, the role of ambassadors, or national differences in professional training affected perceptions. Reflective interpretation would elevate the paper from a descriptive to an explanatory level of analysis.
Underdeveloped Theoretical Framing of Implementation Science Although Leontjevas’ framework is appropriately employed to structure the process evaluation, the broader implementation science literature (e.g., CFIR – Consolidated Framework for Implementation Research; or the RE-AIM model) is not referenced. These established frameworks could help explain variation in adoption, fidelity, and sustainability across contexts.
Recommendation: Integrating concepts such as inner setting (e.g., organisational readiness), outer setting (e.g., national regulations), and intervention characteristics (e.g., perceived complexity or compatibility with existing routines) would provide stronger analytical framing. This could also help better situate findings within the broader field of implementation research in health and social care.
Lack of Statistical Testing The authors deliberately avoid statistical hypothesis testing, citing the descriptive intent of the process evaluation. However, given the availability of Likert-scale and categorical data across a substantial sample (n = 330 for T0 and T6), omitting any statistical comparison limits the strength of the conclusions—particularly when highlighting intergroup differences.
Recommendation: Inclusion of non-parametric tests (e.g., Mann-Whitney U or Kruskal-Wallis for Likert data; Chi-square or Fisher's Exact tests for proportions) would bolster claims about differences between stakeholder types, countries, or intervention groups. Even exploratory inferential analysis would enhance rigour, while clearly signalling that findings are interpretive rather than confirmatory.
Ambiguity in Stakeholder Terminology The classification of stakeholders into "care home managers", "care staff", and "interventionists" is useful but insufficiently nuanced. The manuscript does not clarify how mixed roles or overlaps (e.g., someone acting as both a leisure staff member and informal music leader) were handled analytically. It is also unclear whether role definitions were standardised across countries.
Recommendation: The authors should specify the criteria used for assigning stakeholders to categories and discuss how overlaps were managed. A clear operational table listing roles and associated responsibilities per country or care home type could improve transparency. This is especially important in multinational studies, where job titles and functions can differ significantly.
Sustainability Findings Are Under-analysed A key finding is that most stakeholders were unsure whether the music interventions would be continued after the trial. This has significant implications for the translation of research into practice, yet it is only briefly mentioned in the results and not critically discussed.
Recommendation: The authors should explore potential reasons for low confidence in sustainability (e.g., lack of funding, insufficient training, institutional inertia). They might also consider discussing implementation sustainability in light of broader models such as Nilsen's categorisation of implementation outcomes or Moore et al.’s work on intervention maintenance. Connecting to the literature on "scale and spread" in health and social care interventions could strengthen the manuscript’s impact.
Three extra tips: 1) Expand the discussion of sustainability, particularly regarding implications for future scale-up. 2) Provide clearer definitions and treatment of overlapping stakeholder roles. 3) Proofread the entire manuscript for typographic and syntactic accuracy.
Many thanks once again for submitting this interesting article. I look forward to reading the final version.
Author Response
Thank you for submitting this highly interesting manuscript. The manuscript reports on the process evaluation component of the large-scale, multinational MIDDEL trial, which studied the implementation of music-based interventions (MBIs) for residents in care homes with dementia and depressive symptoms. The study aims to identify barriers and facilitators to MBI implementation by analysing survey responses from stakeholders across five countries. It contributes meaningfully to the literature on complex interventions in geriatric care and music therapy by foregrounding process-related insights in a robust and methodologically transparent manner.
The topic is of high relevance given the global rise in dementia prevalence and the increasing attention to non-pharmacological interventions for neuropsychiatric symptoms in the elderly. The focus on music interventions, which are low-cost and socially engaging, aligns with current trends in person-centred care.
The study design is methodologically robust. Embedding the process evaluation within a cluster-randomised trial and following the framework of Leontjevas et al. ensures theoretical grounding and systematic data collection. The sample size is considerable and covers multiple stakeholder perspectives.
Including data from five countries enhances the study’s generalisability and allows for valuable cross-cultural comparisons in implementation practices and experiences. The dual use of first-order and second-order process data is well-conceived. The integration of quantitative Likert-scale data and binary responses provides a balanced descriptive picture. The inclusion of heatmaps to illustrate intergroup and intragroup differences is highly effective and increases the manuscript’s communicative clarity.
- We are thankful for the positive appreciation of the reviewer regarding our multinational process evaluation. Below we explain how we applied the reviewer's suggestions to improve the manuscript.
There are 5 issues that need extra attention:
Limited Depth in Interpretative Analysis: While the manuscript offers a thorough descriptive overview of stakeholder responses regarding barriers and facilitators, it falls short in exploring why these differences occurred or what they imply for practice and theory. For example, it is reported that certain barriers—such as "high workload" and "lack of motivation"—were anticipated but experienced to a lesser degree than expected. However, the manuscript does not sufficiently unpack potential reasons behind this discrepancy.
Recommendation: The authors could draw on literature about adaptive capacity in care settings or the motivational dynamics of care staff in relation to creative interventions. They might consider whether familiarity with music-based activities, the role of ambassadors, or national differences in professional training affected perceptions. Reflective interpretation would elevate the paper from a descriptive to an explanatory level of analysis.
- Thank you for addressing this issue. Indeed, in our manuscript we aim primarily to provide a descriptive overview of stakeholder responses. Our results are extensive and as a next step (in a follow-up paper) we aim to conduct an in-depth evaluation. Due to the amount of data, we intend to focus only on those results that are hypothesized to be paramount to successful implementation of music-based interventions for people with dementia in care homes. We intend to focus on what is deemed relevant and conduct this in-depth evaluation by linking them to findings from the effect study of the MIDDEL trial. The main effect paper has been submitted for publication elsewhere. We added the following sentence in the discussion as the final sentence: “In a follow-up paper, we intend to synthesize findings of the effect study with findings of the process evaluation. This approach enables us to explore possible discrepancies between expectations and experiences and subsequently relate them to findings of the effect study” (line 673). Nevertheless, if the need for further exploration of these differences in the current manuscript remains and word count/manuscript size allows for it, the authors are willing to revise the manuscript accordingly.
Underdeveloped Theoretical Framing of Implementation Science: Although Leontjevas’ framework is appropriately employed to structure the process evaluation, the broader implementation science literature (e.g., CFIR – Consolidated Framework for Implementation Research; or the RE-AIM model) is not referenced. These established frameworks could help explain variation in adoption, fidelity, and sustainability across contexts.
Recommendation: Integrating concepts such as inner setting (e.g., organisational readiness), outer setting (e.g., national regulations), and intervention characteristics (e.g., perceived complexity or compatibility with existing routines) would provide stronger analytical framing. This could also help better situate findings within the broader field of implementation research in health and social care.
- Thank you for this suggestion. We consulted several papers that incorporated established the abovementioned established frameworks and agree they will be appropriate to help explain variation in adoption, fidelity and sustainability across contexts. We intend to provide this deeper interpretation of implications in a follow-up paper, in which we synthesize process evaluation findings with findings from the effect paper of the main MIDDEL trial, to gain deeper understanding of the role and effect of barriers and facilitators regarding implementation of music-based interventions. We described our motivation to use the Leontjevas framework for the design of the process evaluation in the revised manuscript. In the Discussion section we now added: “The Leontjevas framework was selected over other established frameworks such as the Consolidated Framework for Implementation Research (CFIR) (Damschroder 2022) and the RE-AIM (Reach, Effectiveness, Adoption, Implementation, Maintenance) framework (Holtrop 2021). Leontjevas’ framework fit the aim and scope of our study; is easy to embed in a multinational RCT; and has commonly been applied in care home settings for people living with dementia. A next step in evaluating the implementation of group-based music interventions will be to connect main effect findings with barriers and facilitators for implementation, enabling a translation to day-to-day practice. A combination of elements of the RE-AIM framework and CFIR framework could be appropriate to help systematically explain variation across contexts” (line 597).
Lack of Statistical Testing: The authors deliberately avoid statistical hypothesis testing, citing the descriptive intent of the process evaluation. However, given the availability of Likert-scale and categorical data across a substantial sample (n = 330 for T0 and T6), omitting any statistical comparison limits the strength of the conclusions—particularly when highlighting intergroup differences.
Recommendation: Inclusion of non-parametric tests (e.g., Mann-Whitney U or Kruskal-Wallis for Likert data; Chi-square or Fisher's Exact tests for proportions) would bolster claims about differences between stakeholder types, countries, or intervention groups. Even exploratory inferential analysis would enhance rigour, while clearly signalling that findings are interpretive rather than confirmatory.
- We thank the reviewer for highlighting relevance of statistical testing. Indeed, we have considered this in the development process of the study. The authors decided against it at the time, since response rate was unknown, the survey is not a validated measurement instrument, and the survey consists of a large number of statements. Statistical testing is beyond the focus of the current paper. However, the authors agree that statistical testing can be relevant once we have the outcomes of the effect study. These can direct us to which factors or levels should be investigated further. If differences between countries prove to be particularly relevant, we will, for example, focus on this level and perform statistical tests on it. We are thankful for these detailed suggestions of the reviewer. We look forward to applying these suggestions in the follow-up article in which we connect the effects of music interventions to expected and experienced barriers and facilitators for implementation of said music interventions.
Ambiguity in Stakeholder Terminology: The classification of stakeholders into "care home managers", "care staff", and "interventionists" is useful but insufficiently nuanced. The manuscript does not clarify how mixed roles or overlaps (e.g., someone acting as both a leisure staff member and informal music leader) were handled analytically. It is also unclear whether role definitions were standardised across countries.
Recommendation: The authors should specify the criteria used for assigning stakeholders to categories and discuss how overlaps were managed. A clear operational table listing roles and associated responsibilities per country or care home type could improve transparency. This is especially important in multinational studies, where job titles and functions can differ significantly.
- Thank you for making us aware of this ambiguity. Role definitions were indeed standardized across countries and mixed roles were not applicable. We added this in the manuscript as follows: “This categorization of roles was standardized across countries and overlap between stakeholder roles was not possible” (line 167).
- During survey development researchers from each country discussed terminology of stakeholder roles, job titles and functions, to enable inclusion of all stakeholders in positions as care home managers, care staff, and interventionists. The most common job titles have been described in paragraph 2.3 Participants.
Sustainability Findings Are Under-analysed: A key finding is that most stakeholders were unsure whether the music interventions would be continued after the trial. This has significant implications for the translation of research into practice, yet it is only briefly mentioned in the results and not critically discussed.
Recommendation: The authors should explore potential reasons for low confidence in sustainability (e.g., lack of funding, insufficient training, institutional inertia). They might also consider discussing implementation sustainability in light of broader models such as Nilsen's categorisation of implementation outcomes or Moore et al.’s work on intervention maintenance. Connecting to the literature on "scale and spread" in health and social care interventions could strengthen the manuscript’s impact.
- The authors agree with the reviewer that sustainability is not thoroughly discussed. During the trial the researchers noticed that stakeholders were unaware whether the music interventions would be continued after the six-month intervention period. Several reasons can explain this. First, funding of the research project enabled interventionists to conduct the music interventions on site. When the research funding ended, there were no financial resources available to hire the interventionist. Second, being unaware of continuation could relate to stakeholder role. For example, care staff are essential in facilitating the music session, but do not need to be aware or involved in facilitating continuation of the interventions. Third, since the interventions were initiated by the research institutions involved and not by the care homes themselves, there may not have been a designated person to facilitate continuation of the music interventions. These reflections have been added in the discussion section. We added these reflections in the manuscript as follows: “Low confidence in sustainability may reflect a lack of structural funding, institutional support for long-term integration, or insufficient training and handover mechanisms. Additionally, implementation may have been perceived as a project-based initiative rather than an embedded part of routine care. The majority of respondents were unaware of any plans for sustainable adoption of the MBI after the trial. The uncertainty around sustainability highlights the need to consider long-term implementation outcomes, as described in Nilsen’s framework, where factors such as acceptability, appropriateness, and feasibility need to be aligned with institutional capacity for maintenance (Nilsen, 2015). Future work should consider strategies for scale and spread, which require not only local stakeholder engagement but also alignment with broader system-level levers such as national dementia strategies, professional training structures, and reimbursement pathways” (line 499).
Three extra tips: 1) Expand the discussion of sustainability, particularly regarding implications for future scale-up. 2) Provide clearer definitions and treatment of overlapping stakeholder roles. 3) Proofread the entire manuscript for typographic and syntactic accuracy.
Many thanks once again for submitting this interesting article. I look forward to reading the final version.
- We thank the reviewer for these extra tips and have applied them to the revised manuscript accordingly.
Reviewer 3 Report
Comments and Suggestions for Authors
Thank you for submitting this paper for publication consideration. This project is important and the authors have done a thorough job on the paper. I only have a couple of suggestions and these are really only copy edits.
Abstract
- Recommend making barriers and facilitators read the same (i.e., use of colon). Please add "and" before "lack of motivation" (same with 4.1 in discussion)
Literature review
- Line 75 - please edit to MBIs
Discussion
- Line 483 - recommend adding "The" so it reads "The majority of..."
- Line 610 - suggest avoiding ending sentence with preposition (to)
- Perhaps note that cost effectiveness studies/2nd order evaluation are necessary based on the Leontjevoas framework recommendations. This was noted earlier and, as the authors are well aware, is an important necessary next step for future research.
I wish the authors well and look forward to reading more from them in the future.
Author Response
Thank you for submitting this paper for publication consideration. This project is important and the authors have done a thorough job on the paper. I only have a couple of suggestions and these are really only copy edits.
- We thank the reviewer for acknowledging the importance of our process evaluation. In our response below we clarify how we incorporated the suggestions provided by the reviewer.
Abstract: Recommend making barriers and facilitators read the same (i.e., use of colon).
- As suggested by the reviewer, we revised this accordingly: “Barriers include: lack of support, changes in employees, lack of motivation. Facilitators include: people involved facilitate and stimulate implementation, stable well-functioning teams, clear communication, and maintaining project planning” (line 31).
Please add "and" before "lack of motivation" (same with 4.1 in discussion)
- We added “and” before “lack of motivation” in the abstract and in paragraph 4.1 in the Discussion section.
Literature review: Line 75 - please edit to MBIs
- Based on another reviewer’s suggestion, we rephrased this sentence in its entirety: “While evidence for barriers and facilitators for MBIs are known, this remains limited for group MBIs” (line 75).
Discussion: Line 483 - recommend adding "The" so it reads "The majority of..."
- We added “The”.
Line 610 - suggest avoiding ending sentence with preposition (to)
- We rewrote it as follows: “In future trials, a deliberate decision can be made whether or not to remind respondents at the start of the survey to which arm their CHU was allocated” (line 654).
Perhaps note that cost effectiveness studies/2nd order evaluation are necessary based on the Leontjevoas framework recommendations. This was noted earlier and, as the authors are well aware, is an important necessary next step for future research.
- The authors thank the reviewer for addressing this. We added this as an important and necessary next step for future research in paragraph 4.3 in line 662, as follows: “For a complete picture of sustainable implementation, assessment of cost-effectiveness of the MBIs will be assessed”.
I wish the authors well and look forward to reading more from them in the future.
Reviewer 4 Report
Comments and Suggestions for Authors
Thanks for the opportunity to review this paper about a multinational review of barriers and facilitators for implementing music-based interventions in care homes.
Some comments for consideration:
Abstract line 31 - What is MBIs? Is it music-based interventions? This should be made clearer.
Abstract line 32 - For symmetry, should the "facilitators include:" not have the : just like barriers? Also, it may be better to phrase as "people involved facilitating and stimulating implementation"
Abstract lines 34-39 needed several reads to understand. Could these be rephrased?
Introduction lines 45-46 - This sentence is long and not easy to read. Can it be broken into shorter sentences?
In general, the Introduction may benefit from some reorganisation and rephrasing. For example, 2nd paragraph - This paragraph was confusing to read. For example, it is not clear how costs could be an implementation strategy but later mentioned as a barrier "high costs". The flow was hard to follow too. Would it be possible to add a topic sentence e.g., "While evidence for barriers and facilitators for MBIs are known, this remains limited for group MBIs. For MBIs..."?
Another example, the last paragraph of the introduction has a long sentence with many "and"s in lines 91-94. This needed repeated reading to understand.
The methods and results section are easier to follow. For the strengths and limitations section, can it be reworded as "This study has several strengths" and "This study also has several limitations"? This would make it easier to read.
Overall, this process evaluation is very necessary. I feel this interesting study could be made easier to read with some reorganisation and rephrasing.
Author Response
Thanks for the opportunity to review this paper about a multinational review of barriers and facilitators for implementing music-based interventions in care homes. Some comments for consideration:
- We kindly thank the reviewer for their suggestions. In our response below we clarify how we incorporated the comments in the manuscript.
Abstract line 31 - What is MBIs? Is it music-based interventions? This should be made clearer.
- We added the abbreviation MBIs in parentheses at its first mention, at line 28.
Abstract line 32 - For symmetry, should the "facilitators include:" not have the : just like barriers? Also, it may be better to phrase as "people involved facilitating and stimulating implementation"
- Thank you for point this out. We revised this accordingly: “Barriers include: lack of support, changes in employees, lack of motivation. Facilitators include: people involved facilitating and stimulating implementation, stable well-functioning teams, clear communication, and maintaining project planning” (line 31).
Abstract lines 34-39 needed several reads to understand. Could these be rephrased?
- We thank the reviewer for addressing this. In line 38 the word “of” has been removed, as follows: “[…] yet implementation of in care homes can be challenging due to contextual factors”. The sentence “MBIs can be beneficial for people with dementia, yet implementation in care homes can be challenging due to contextual factors” has been moved up, so that it fits better with the sentences that follow (line 35).
Involving stakeholders in key positions is essential: care home managers are pivotal for policy making and sustainable adoption of MBIs, whereas commitment and involvement of care staff is needed for day-to-day implementation. Insight into influential barriers and facilitators of implementation can contribute to the interpretation of effect study results.
Introduction lines 45-46 - This sentence is long and not easy to read. Can it be broken into shorter sentences?
- In line with the reviewer’s suggestion, we broke this sentence into two shorter sentences by removing “and”.
In general, the Introduction may benefit from some reorganisation and rephrasing. For example, 2nd paragraph - This paragraph was confusing to read. For example, it is not clear how costs could be an implementation strategy but later mentioned as a barrier "high costs".
- We thank the reviewer for pointing out issues of readability and providing specific examples for improvement. We accommodated this by replacing the word 'include' with 'relate to', so that examples of strategies and factors described in recent literature follow more logically: “Common implementation strategies of MBIs for people with dementia in care homes relate to education, quality management, planning, restructuring, and costs” (line 65).
The flow was hard to follow too. Would it be possible to add a topic sentence e.g., "While evidence for barriers and facilitators for MBIs are known, this remains limited for group MBIs. For MBIs..."?
- Thank you for this elaborated example. To improve the flow, we placed this sentence at line 75, replacing the previous sentence, as follows: “While evidence for barriers and facilitators for MBIs are known, this remains limited for group MBIs” (line 75).
Another example, the last paragraph of the introduction has a long sentence with many "and"s in lines 91-94. This needed repeated reading to understand.
- Thank you for pointing out difficulty reading this section. We split up the long sentence in several shorter sentences, as follows: “The primary objective of the process evaluation was to gain insight into first-order process data and second-order process data. First-order process data include: intervention quality, measured by satisfaction, relevance and feasibility. Second-order process data include: presence and influence of anticipated and experienced potential barriers and facilitators for implementation of MBIs for people with dementia and depression in care homes of different stakeholders” (line 678).
The methods and results section are easier to follow. For the strengths and limitations section, can it be reworded as "This study has several strengths" and "This study also has several limitations"? This would make it easier to read.
- In line with the reviewer’s suggestion, we changed “knows” to “has”.
Overall, this process evaluation is very necessary. I feel this interesting study could be made easier to read with some reorganisation and rephrasing.
- We appreciate the positive evaluation of the reviewer regarding this process evaluation. We reorganized and rephrased sections based on feedback of all reviewers and hope this appropriately improved the manuscript.
Round 2
Reviewer 1 Report
Comments and Suggestions for Authors
The majority of the requested revisions have been thoroughly and appropriately incorporated. I appreciate your thoughtful work.